# Morphological and Ultrastructural Features of Selected Epidendroideae Pollen Dispersal Units and New Insights into Their Chemical Nature

**DOI:** 10.3390/plants13081114

**Published:** 2024-04-16

**Authors:** Carola Purgina, Silvia Ulrich, Martina Weber, Friðgeir Grímsson

**Affiliations:** 1Department of Botany and Biodiversity Research, Division of Structural and Functional Botany, University of Vienna, 1030 Vienna, Austria; silvia.ulrich@univie.ac.at (S.U.); martina.weber@univie.ac.at (M.W.); 2Department of Historical Archaeology, Austrian Archaeological Institute (OeAI), Austrian Academy of Sciences (OeAW), 1010 Vienna, Austria

**Keywords:** orchids, epidendroids, elastoviscin, electron microscopy, pollen dispersal units, pollen tetrads, pollinarium, pollinium, sporopollenin

## Abstract

Orchidaceae display enormous diversity in their flower morphology, which is particularly evident in their pollen dispersal units (pollinia, pollinaria). The packaging of pollen by elastoviscin leads to a great diversity of these morphologically and structurally complex pollen units. Despite being one of the most diverse angiosperm families, the available palynological data on orchids remain limited and sometimes contradicting. This study provides new insights into the pollen morphology and ultrastructure of five orchid species from the subfamily Epidendroideae, using combined light, scanning electron, and transmission electron microscopy. The aim was to compare the morphology and ultrastructure of pollen dispersal units and to elucidate the chemical nature of the pollen wall layers and of elastoviscin. Our combined light and electron microscopy investigation demonstrated the presence of six tetrad types even within a single pollinium, which is unique for orchids. The application of different staining methods confirmed the assumed lipidic nature of elastoviscin and the differences in its contrast and ultrastructure suggest a mixture of sticky materials with dissimilar chemical compositions. This study affirmed that sporopollenin is mostly restricted to the outer pollen grains of peripheral tetrads in compact and sectile pollinia, while inner tetrads exhibit highly reduced non-sporopollenin pollen walls.

## 1. Introduction

### 1.1. Orchids and Their Pollen Dispersal Units

The Orchidaceae family is one of the largest and most highly evolved monocot families among angiosperms [1,2,3], with an estimated number of species ranging from 24,190 [4] to 30,000 [5,6], grouped into 736 [7] to 1000 [8] genera. Chase et al. [7] and Li et al. [9] classify orchids into five subfamilies: Apostasioideae Horan., Cypripedioideae Kostel., Epidendroideae Kostel., Orchidoideae Eaton, and Vanilloideae (Lindley) Szlachetko. Orchid flowers, across all subfamilies, exhibit extreme flower synorganization, characterized by structures such as the gynostemium (column), which bears the stigma and anther(s) that contain pollen. To elucidate their complex pollen dispersal units (PDUs), the illustrations in Figure 1 depict all types and parts of the accessory structures (the caudicle, stipe, and viscidium). The PDUs in Orchidaceae range from monads or loose/permanent tetrads in the basal subfamily Apostasioideae to agglutinated pollen tetrads (that can disintegrate) in Vanilloideae and Cypripedioideae, and permanent pollinia (mainly composed of permanent tetrads) in Orchidoideae and Epidendroideae [4]. When the pollinium is attached to appendices (like a caudicle, stipe, or viscidium), it is called a pollinarium [10] (Figure 1B–E). The number and shape of the pollinia per theca vary among subfamilies, ranging from two, four, or six to eight, depending on the structure of the septum in each theca, which either separates the sporogenous tissue or unites the contents of distinct locules [2,11,12]. Prior to anthesis, the agglutination of the pollen content from two thecae (the connection of two bipartite pollinia) with the sticky material elastoviscin results in either a longitudinal, dorsiventral, or lateral line in the mature pollinium, referred to as a suture [11,13]. Pollinia with a suture occur only in a few tribes, such as Vandeae Lindl. and Cymbidieae Pfitzer of the Epidendroideae (also known as epidendroids) and Orchideae Verm. of the Orchidoideae [13]. Depending on the degree of the agglutination of the monads/tetrads forming the pollinium/pollinia, as well as the septation of the sporogenous tissue within the thecae, the pollen packages can be categorized as soft, hard (the highest degree of cohesion), or sectile [12,13] (Figure 1). Sectile pollinia consist of tetrads agglutinated into massulae (Figure 1D), which are attached at the base by elastoviscin. Pollen grains are arranged in different tetrad formations within a pollinium and a massula (in the case of a sectile pollinium) [1,3,14]. Data on the number of pollen grains per pollinium vary from 5000 to 4,000,000 depending on the pollen package type [12,15,16]. Soft pollinia exhibit the lowest number of pollen grains, sectile pollinia an intermediate number, and hard pollinia the highest [15]. 

### 1.2. Morphological Adaptation to Pollinators—The Types and Roles of the Accessory Structures (Caudicle, Stipe, and Viscidium)

Pollinia are frequently attached to appendages, forming pollinaria (Figure 1B–E). The caudicle is an extension of the pollinium/pollinia, originating from the sporogenous tissue in the anther [12,17] (Figure 1C,E), with its structure ranging from hard (containing mainly pollen grains) to elastic [18]. Elastic caudicles are primarily composed of elastoviscin, giving them a transparent and shiny appearance [18]. Depending on its composition, the caudicle can be either appendicular (adherent) or frenicular (elastic band-like) [13,19] (Figure 1C,E). An appendicular caudicle consists of scattered tetrads (mostly abortive) embedded in sticky material, while a frenicular caudicle comprises sticky materials only and lacks pollen grains/tetrads [11,12,13,17]. The appendicular caudicle has a weak point, facilitating the separation of the pollinium from the pollinator during pollination [12,17,18]. The frenicular caudicle connects the pollinium/pollinia to the stipe, which also possesses a break point during pollination [19] (Figure 1E). Stipes derived from columnar tissue (the rostellum) are rare outside of the higher epidendroids (vandoids) (incl. the tribes Calypsoeae Dressler, Cymbidieae Pfitzer, Maxillarieae Pfitzer, and Vandeae Lindl.) (e.g., [12,13,17] and the references therein). The stipe is a non-viscid, cellular stalk or strap connecting the pollinium/pollinia to the viscidium [4,14,17] (Figure 1E). The viscidium is a removable, often pad-like structure derived from the rostellum [4,20] (Figure 1B–E), and it releases glue-like materials following the breakdown of stigmatic cells, allowing the viscidium to become sticky. This structure is crucial for efficient pollination as it adheres the pollinarium to the pollinator [12,17]. 

### 1.3. Elastoviscin and Its Manifold Functions

Elastoviscin is a highly viscous, non-acetolysis-resistant substance produced in the cytoplasm of tapetal cells [14] in all subfamilies of Orchidaceae [21,22]. This sticky material has various functions and places of origin within the anther: it agglutinates monads/tetrads; coheres massulae at their base, forming a sectile pollinium; constitutes a primary component of caudicles (appendicular and frenicular); attaches pollinium/pollinia to stipes; is present in cells of the viscidium; and adheres monads/tetrads/pollinia/pollinaria to pollinators [14]. Despite its importance, the chemical composition of elastoviscin remains poorly understood, and, so far, only unsaturated fatty acids have been identified as one of its main components [14,23].

### 1.4. Orchid Pollen’s Morphology and Ultrastructure—The Status Quo

Pollen morphology in Orchidaceae exhibits remarkable diversity, evident not only at the macroscopic level (e.g., number of pollinia, packaging of pollen) but also at the microscopic level, including the pollen’s wall structure and ornamentation [11,24,25]. The ornamentation of orchid pollen extends over the entire known breadth of pollen, ranging from psilate, reticulate-heterobrochate, striate-rugulate, and foveolate to fossulate, displaying shared features within subfamilies [24,26,27]. The aperture configuration in orchid pollen can be sulcate, ulcerate, porate, or inaperturate [3,28]. Ultrastructural studies of orchid pollen walls (e.g., [24,29,30,31,32,33]) have revealed greater complexity and differences compared to the “classical” tectate-columellate pollen walls observed in most angiosperm pollen (e.g., [10,34,35]). The interpretation of individual pollen wall layers and the application of the appropriate terminology based on comparative observations are challenging due to the limited studies on orchid pollen/pollinia that use combined methods [24,36]. However, the orchid pollen wall can generally be differentiated into two layers: a sporopollenin-based exine (with the endexine absent) and a polysaccharidic intine [24,28,29]. The intine is present in every monad/tetrad of the pollinium/massula and is typically either monolayered or, more commonly, two- to three-layered [29]. In case of both compact and sectile pollinium/pollinia, the exine is primarily limited to peripheral tetrads [23,24,29]. In Epidendroideae, the exine structure varies from tectate to semitectate, with mostly columellate but occasionally globular infratectum or without infratectum and/or a foot layer [24,37], and, within the pollinia/massulae, its sporopollenin elements decrease in size and quantity inwardly [24]. In the inner tetrads, the exine is usually absent [24,29]. 

### 1.5. The Purpose of This Study

Despite the popularity, diversity, and species richness of orchids, palynological studies on this family remain rare. The research conducted so far has predominantly focused on scanning electron microscope (SEM) investigations (e.g., [12,26,27]), with limited attention given to light microscope (LM) (e.g., [1,3,13]) or transmission electron microscope (TEM) (e.g., [24,29,32]) studies. Notably, combined LM/SEM/TEM investigations are completely lacking. Thus, the primary aim of this study was to provide a comprehensive analysis of both the morphological and ultrastructural features of pollinia/tetrads/pollen and the chemical composition of the pollen walls, from selected orchid species of the higher epidendroids (*Oncidium crocidipterum* (Rchb. f.) M.W.Chase & N.H.Williams and *Polystachya cultriformis* (Thouars) Lindl. ex Spreng.) and lower epidendroids (*Bulbophyllum retusiusculum* Rchb. f., *Dendrobium* × *delicatum* (F.M. Bailey) F.M. Bailey, and *Cephalanthera longifolia* (L.) Fritsch). To achieve this, we used various light microscopic techniques (white and fluorescence light) and staining methods (Toluidine blue, acetocarmine, potassium iodine) to assess the differences in autofluorescence, cellular condition, pollen wall features, and starch content among their pollinia/tetrads/pollen. Additionally, we applied different contrast methods for TEM to elucidate the chemical composition of individual pollen wall layers. Through these approaches, we aimed to significantly expand the palynological data concerning Orchidaceae pollen, resolving existing discrepancies and providing new insights into the ultrastructure of the orchid pollen wall and the chemical nature of the “middle” pollen wall layer and elastoviscin. 

We hypothesized that the selected orchid species from different genera would exhibit morphological variability in their pollen dispersal units (PDUs), including differences in size and shape, but not in ornamentation. We also anticipated that, contrary to lower epidendroids, all investigated higher epidendroid species would exhibit a pollinarium with a stipe and frenicular caudicle. Regarding their ultrastructure, we expected the chemical nature of the middle layer in all epidendroids to be similar, despite conflicting descriptions in the literature. Based on the literature, we presumed that most studied orchid species possess permanent tetrads, as well as various types of tetrads. Combined LM/SEM/TEM investigations are time-consuming and also depend strongly on the availability of fresh material. Therefore, our selection of orchid species considered their flowering period within the time frame of the study, as well as the morphological differences in their PDUs and the presence of permanent tetrads vs. monads. Overall, the inclusion of both lower and higher epidendroids in our study facilitates a comprehensive investigation of the morphological, ultrastructural, and chemical aspects of orchid pollinia/tetrads/pollen, contributing to a deeper understanding of these complex orchid pollen dispersal units.

## 2. Results

### Systematic Description of Epidendroideae Pollinium/Tetrads/Pollen

The findings for the investigated Epidendroideae species are presented according to their subdivision into lower and higher epidendroids and arranged alphabetically by genera. Each species is described in terms of the morphology and size of its pollinarium/pollinium, tetrad types, and the size of its single pollen grains, initially based on LM studies. Subsequently, the ornamentation and the ultrastructure of the pollen wall, and its cell content, are delineated based on SEM and TEM. The pollen terminology used follows Halbritter et al. [10].

Description of species *Bulbophyllum retusiusculum* Rchb. f. (Figure 2, Figure 3, Figure 4, Figure 5, Figure 6 and Figure 7; Table 1, Table 2 and Table 3), lower epidendroids: 

The pollen dispersal unit of *Bulbophyllum retusiusculum* comprises a compact bean-shaped pollinium (Figure 2B and Figure 3A) measuring 0.69 mm in length and 0.46 mm in width (LM), showing autofluorescence (Figure 2C, Table 1). The pollinium is composed of tetrads (Figure 2D–I and Figure 3B) glued together by pollen coatings (Figure 3C,D). Various types of tetrads have been identified, including tetrahedral, planar-tetragonal, planar-rhomboid, and decussate (Figure 2D–G). In LM, the hydrated pollen grains within tetrads range in size from 10.8 µm to 14.3 µm (Table 1). Mature pollen grains are inaperturate and two-celled at anthesis (Figure 2H and Figure 7A). Toluidine blue staining of the tetrads revealed differences in the pollen wall structure (Figure 2E).

**Figure 3 plants-13-01114-f003:**
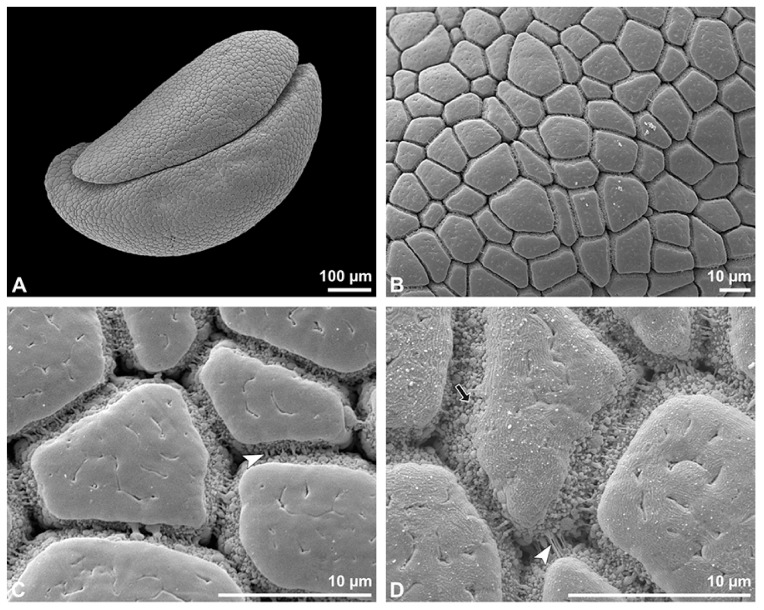
SEM micrographs of *Bulbophyllum retusiusculum* pollinium/pollen. Note: (**A**) Overview of hydrated (DMP + CPD) pollinia. (**B**) Tetrad arrangement. (**C**) Tetrads with psilate, perforate to fossulate ornamentation, and pollen coatings in between (white arrowhead). (**D**) Exine of outer tetrads; perforate, fossulate, granulate (black arrow), with fine striate ornamentation and pollen coatings (white arrowhead) connecting the tetrads.

Under LM and SEM, the outermost tetrads constitute a thick pollen wall (Figure 2F) with psilate to fine striate to fossulate ornamentation and perforations (Figure 3C,D). Contrary to the outer thick-walled tetrads of the pollinium, the inner tetrads have only thin pollen walls (Figure 2G). 

Ultrastructural studies reveal a trilaminate structure to the pollinium, consisting of outer tetrads with thick pollen walls and detached protoplasts (stratum 1), tetrads with detached protoplasts and reduced pollen walls (stratum 2), and inner tetrads with reduced pollen walls (stratum 3) (Figure 4 and Figure 5A,B). The peripheral part of the pollinium (stratum 1) shows tetrads with pollen walls comprising three layers (Figure 5C,D). The outermost layer (layer 1), a continuous, compact sporopollenin layer (Figure 5C,D), constitutes a common pollen wall layer on the distal half of the outer pollen grains only (Figure 5C). In the contact zones of neighbouring outer tetrads of the pollinium, the compact sporopollenin layer transitions into smaller discontinuous subunits (Figure 5C). A subjacent electron-translucent pollen wall layer (layer 2) surrounds each pollen grain (monad), forming a common layer around each tetrad (Figure 5C,F). Layer 2 is much thicker than layer 1 and comprises electron-dense columellate to granulate elements (Figure 5D). It decreases inwardly in thickness and density (Figure 5C). The chemical composition of layer 2 appears to be polysaccharidic, as staining with potassium permanganate does not indicate that it has a lipidic nature (Figure 6G,H). The thin innermost wall (layer 3) is monolayered and most likely an intine. At the distal half of the monads, this layer is undulating due to the detachment of the generative cell (Figure 5E,H). The pollen walls of the inner pollen grains of the outer tetrads are reduced and consist only of layer 2 and layer 3 (intine) (Figure 5E). The protoplasts of all outer tetrads of stratum 1 are detached from the intine, have no intact nuclei and are poor in organelles (Figure 4 and Figure 5C), suggesting infertility. In the contact area of two pollinia (Figure 4) the tetrads did not show detached protoplasts, indicating the fertility of the mature two-celled pollen grains (Figure 5F). At the contact area of the outer pollen grains of the tetrads, the pollen wall layers 2 and 3 are slightly thinner (Figure 5G). In stratum 2, the pollinium exhibits tetrads with highly reduced pollen walls (Figure 6A), consisting of layer 2 and layer 3 only (Figure 6B). Similar to the tetrads of stratum 1, layer 2 surrounds each tetrad and forms the outer pollen wall layer of each monad (Figure 6B). The inner layer (layer 3), a monolayered intine, is undulating in the distal half of the monads (Figure 6C). As observed in the outer tetrads of stratum 1, the protoplasts are detached and poor in organelles (Figure 4 and Figure 6A). Tetrads of the inner part of the pollinium (stratum 3) resemble those of stratum 2, with reduced pollen walls consisting of layer 2 and intine (layer 3) (Figure 6D–F). Contrary to the tetrads of stratum 1 and stratum 2, the tetrads of stratum 3 contain protoplasts richer in organelles and without a detached protoplast (Figure 4 and Figure 6D). Pollen reserves like starch are absent in the tetrads of all three strata (Figure 2I); only some lipid droplets are present (Figure 5F). Some tetrads of stratum 2 and 3 exhibit plasma bridges between the monads (Figure 6E), suggesting that these tetrads have not reached their final mature stage, where plasma bridges are usually entirely closed. The spaces between the tetrads of all strata are filled with an electron-translucent filamentous material (Figure 7B) gluing the tetrads together.

Remarks: To date, there are only few pollen studies on this genus, although it comprises c. 2168 species [38]. Schill and Pfeiffer [24] conducted SEM studies on nine species of *Bulbophyllum*, describing a predominantly psilate ornamentation, which is in accordance with our results for *B. retusiusculum*. Zavada [39] has provided only TEM work to date, which has highlighted the structural differences in the pollen wall of *B. imbricatum* Lindl. between the peripheral and inner tetrads. Our study on *B. retusiusculum* corroborates these findings, and additionally shows differences in the protoplasts’ conditions within tetrads, depending on their position within the pollinium. The peripheral tetrads of stratum 1 exhibit sporopollenin wall elements and detached protoplasts, while tetrads from the centre (stratum 3) display reduced pollen walls without sporopollenin elements and intact protoplasts. Notably, the pollen wall structure deviates from the classical pollen wall model (see [10]), particularly in the presence of the unique layer 2, which is not comparable to any known wall layer. Additionally, layer 3 deviates from a classical intine, appearing more like a primary cell wall.

Description of species *Cephalanthera longifolia* (L.) Fritsch (Figure 8, Figure 9, Figure 10, Figure 11 and Figure 12; Table 1, Table 2 and Table 3), lower epidendroids: The pollen dispersal unit constitutes a soft banana-shaped pollinium (Figure 9A) with an average length of 3.2 mm and an average width of 0.3 mm that shows autofluorescence in LM (Figure 8B,C, Table 1). It consists of tetrads (Figure 8D and Figure 9B) agglutinated by pollen coatings (Figure 8H). These tetrads easily disintegrate into heteropolar monads, displaying, in LM, a size ranging in their hydrated condition between 26.4 µm and 33.1 µm (Figure 8I, Table 1). Monads are ulcerate, spheroidal, isodiametric to oblate, and heteropolar (Figure 8E,F) and have a suboblate P/E ratio in SEM (Figure 9D). Within a tetrad, the ulcus is positioned at the proximal pole and is not visible in the intact pollinium (Figure 9B). 

In SEM, the pollen wall ornamentation in the interapertural area is foveolate to reticulate-heterobrochate, with thick muri (Figure 9C). The size of the lumina ranges from 1.14 µm to 3.31 µm (mean 2.16 µm). The aperture membrane is ornamented (nano- to micro-verrucate, granulate) (Figure 9D).

**Figure 4 plants-13-01114-f004:**
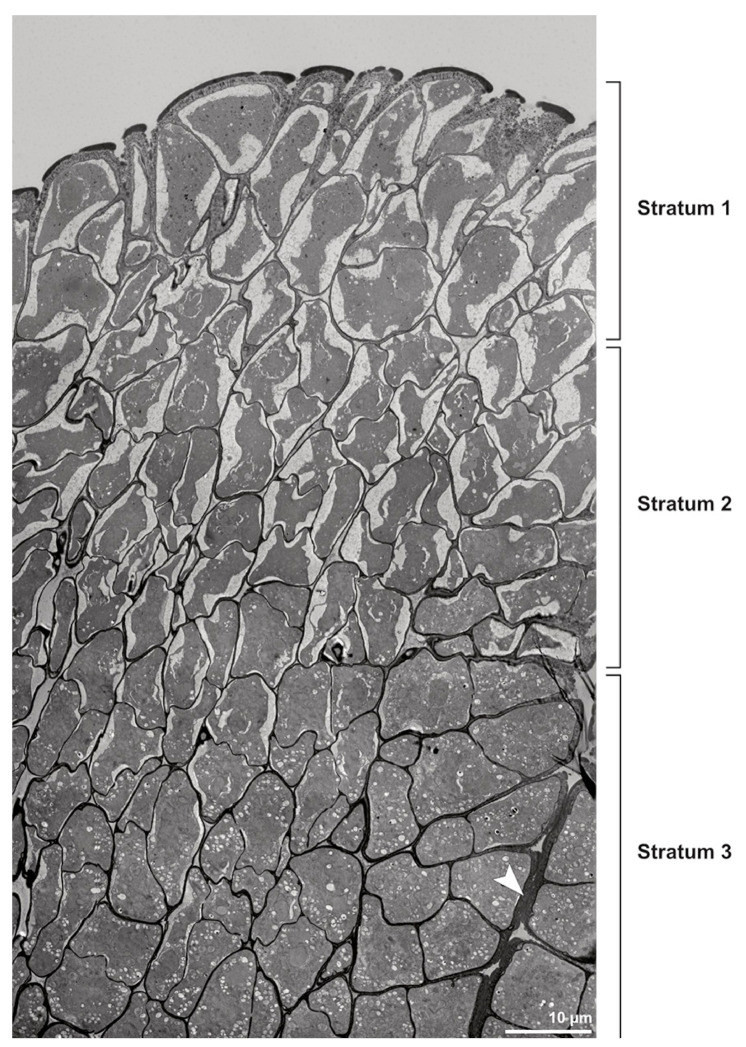
TEM panorama micrograph showing a cross-section of two adjacent *Bulbophyllum retusiusculum* pollinia. Note: The pollinia can be divided into three strata based on variations in the tetrad/pollen wall structure and pollen protoplast ultrastructure. This stratification can be traced from the peripheral margin (stratum 1) inwards (stratum 2 and 3). The white arrowhead indicates contact zone of the two pollinia.

**Figure 5 plants-13-01114-f005:**
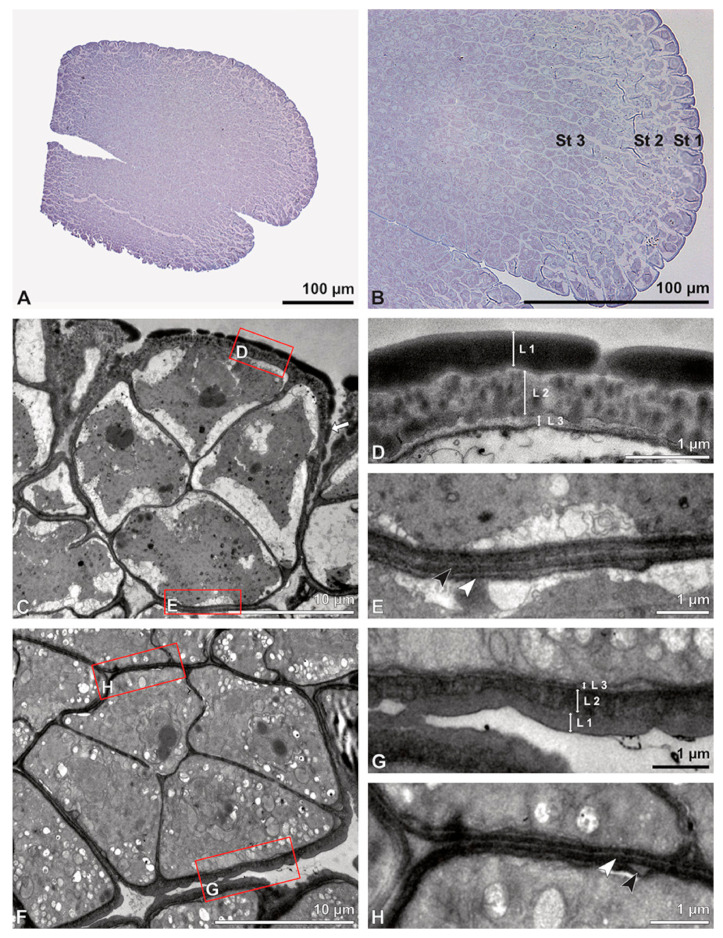
LM (**A**,**B**) and TEM (**C**–**H**) micrographs showing cross-sections of *Bulbophyllum retusiusculum* pollinia/tetrads. Note: (**A**,**B**) Semi-thin section of two adjacent pollinia showing different strata (St 1, St 2, and St 3), TBO. (**C**) Tetrad of outer stratum 1 with detached protoplasts and a three- (**D**) to two-layered (**E**) pollen wall; the white arrow points to discontinuous sporopollenin subunits. (**D**) Detail of pollen wall, showing three layers (L1, L2, and L3). (**E**) Detail of pollen wall consisting of L2 (black arrowhead) and L3 (intine), with detached undulating areas (white arrowhead). (**F**) Tetrad of outer stratum 1 close to an adjacent pollinium. (**G**) Detail of pollen wall, showing three different layers (L1, L2, and L3). (**H**) Detail of pollen wall, showing L2 (white arrowhead) and L3 (black arrowhead).

**Figure 6 plants-13-01114-f006:**
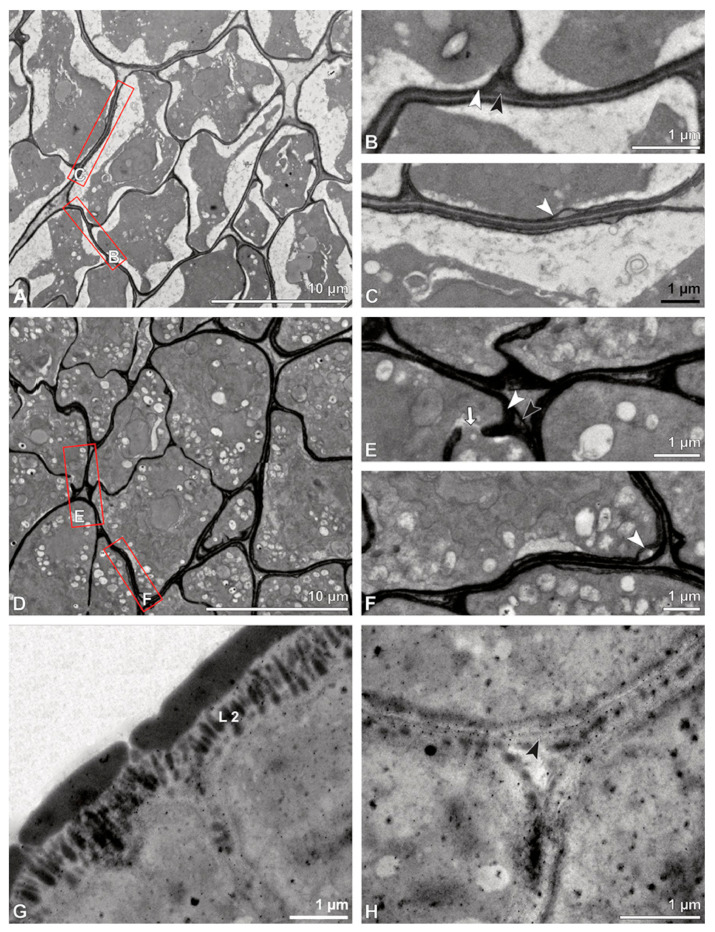
TEM micrographs showing cross-sections of *Bulbophyllum retusiusculum* pollinia/tetrads. Note: (**A**) Tetrad of stratum 2 with detached pollen protoplasts and two-layered pollen wall. (**B**) Detail of a pollen wall with two layers: L2 (black arrowhead) and L3 (intine, white arrowhead). (**C**) Detail of a pollen wall with a distally undulating intine (L3, white arrowhead). (**D**) Tetrad of the innermost stratum 3. (**E**). Detail of pollen wall consisting of L2 (black arrowhead) and L3 (white arrowhead) and plasmodesmos (white arrow). (**F**) Detail of a pollen wall showing the distally undulating intine (L3, white arrowhead). (**G**) Pollen wall of outer tetrad of stratum 1 with an electron-translucent L2, KMnO_4_. (**H**) Contact zone of three tetrads from stratum 3; the black arrowhead points to L2, KMnO_4_.

**Figure 7 plants-13-01114-f007:**
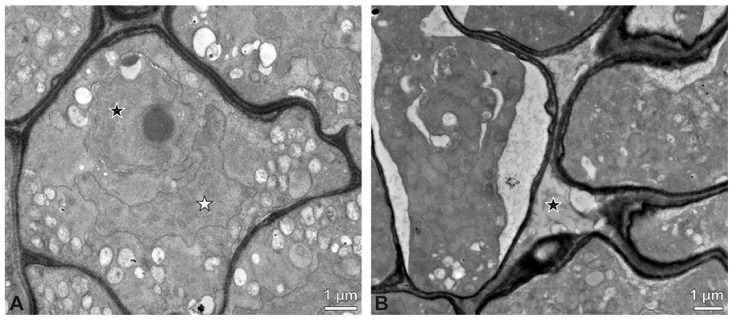
TEM micrographs showing cross-sections of *Bulbophyllum retusiusculum* pollinia/tetrads. Note: (**A**) Mature two-celled pollen grain with generative cell (black star) and vegetative nucleus (white star). (**B**) Filamentous material between tetrads (black star).

A combined study of the pollinium using LM and TEM shows a loose agglutination of tetrads/monads by pollen coatings (Figure 10 and Figure 11A,B). The pollen wall structure of each monad is composed of an ektexine, layer 2, and an intine. The ektexine is subdivided into a discontinuous tectum, columellate infratectum, and thin, continuous foot layer (Figure 11C,D). At the transition from the interapertural area to the aperture, the foot layer becomes undulating and imbedded in layer 2. An endexine is not present (Figure 11F,G). The subjacent electron-translucent wall layer 2 and the thin electron-dense intine surround the monads and increase in thickness towards the aperture region (Figure 11C–E). Layer 2 increases in thickness towards the aperture and shows lamellated electron-dense interstratifications adjacent to the foot layer (Figure 11C–E). At anthesis, the monads are two-celled (Figure 8G and Figure 12A). Reserves in the cytoplasm, like lipids or starch, are absent (Figure 8H and Figure 12A). Two different types of pollen coatings are present: electron-dense pollenkitt in the infratectum (Figure 11D and Figure 12B) and an additional electron-dense material between the monads (Figure 12B).

Remarks: The pollinia of *Cephalanthera longifolia* and two other species of this genus (*C. damasonium* (Mill.) Druce, *C. rubra* (L.) Rich.) have been investigated using TEM by Barone Lumaga et al. [32]. These authors concluded that the ulcerate aperture of *C. longofolia* has a distal position and that there is an endexine layer comprising part of the pollen wall. This contradicts our results, which clearly show the aperture to be in a proximal position and that the proclaimed endexine is most likely an intine. Contrary to other species examined in this study, *C. longifolia* has a soft pollinium composed of tetrads that disintegrate easily into monads. Also, the structure of the pollen wall is the same regardless of the internal position of the pollen grains within the pollinium. Furthermore, the pollen wall in *C. longifolia* is more comparable to the “typical” tectate-collumelate pollen wall of other angiosperms, as portrayed in Halbritter et al. [10]. *Cephalanthera longifolia* is the only species of those examined herein that has a reticulate ornamentation and an aperture.

Description of species *Dendrobium × delicatum* (F.M. Bailey) F.M. Bailey (Figure 13, Figure 14, Figure 15, Figure 16, Figure 17 and Figure 18; Table 1, Table 2 and Table 3), lower epidendroids: The dispersal unit is an elliptical-shaped pollinium 1.5 mm in length and 0.5 mm in width (SEM) (Figure 14A, Table 1). It consists of tetrads (Figure 14B) adhered together by pollen coatings (Figure 14D). Various tetrad types were observed within the pollinium: tetrahedral, planar-tetragonal, planar-linear, and decussate (Figure 13B). Under LM, the size of the hydrated pollen grains within the tetrads ranges from 12.6 µm to 15.9 µm (Table 1). The monads are inaperturate and two-celled at anthesis (Figure 13D and Figure 17G). Staining with basic fuchsin in LM revealed differences in the pollen wall structure of the tetrads, depending on their position within the pollinium (Figure 13C). 

**Figure 8 plants-13-01114-f008:**
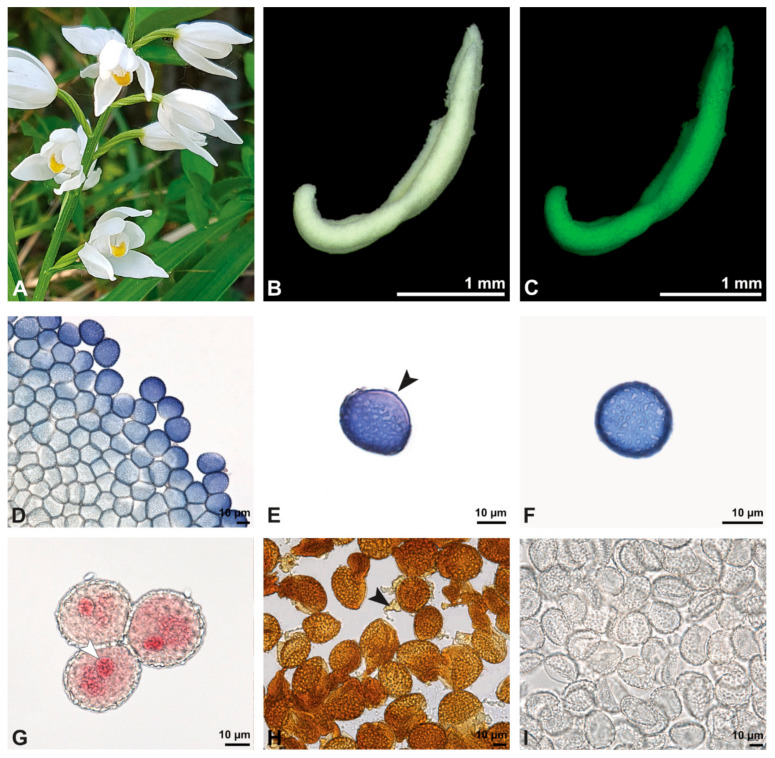
Macroscopic image of inflorescence/flowers (**A**) and LM micrographs (**B**–**I**) of *Cephalanthera longifolia* pollen. (**B**) Pollinium. (**C**) Pollinium fluorescing under UV light. (**D**) Arrangement of monads within pollinium, TBO. (**E**) Ulcerate (black arrowhead) pollen grain from an equatorial view, reticulate ornamentation in interapertural area, TBO. (**F**) Pollen grain from a polar view, TBO. (**G**) Binucleate pollen and its generative cell (white arrowhead), Carmine. (**H**) No starch detected; pollen coatings connect tetrads (black arrowhead), Lugol. (**I**) Hydrated reticulate pollen.

In SEM, the outermost tetrads exhibit thick wall elements with an outer layer displaying either a psilate, perforate to fossulate (Figure 14D), or granulate ornamentation (Figure 14C). Compared to the outer tetrads, the inner tetrads consist of a thin pollen wall only (Figure 13C).

These variations are also clearly visible in TEM, which revealed differences in the cell content as well. Based on this combined study, the pollinium can be divided into three distinct strata, from the peripheral margin to the inside (Figure 15). Outermost thick-walled tetrads with detached protoplasts characterize the outer stratum 1. In the adjacent stratum 2, the tetrads are thin-walled and the protoplasts are also detached, whereas the inner tetrads of stratum 3 are also thin-walled but with intact protoplasts (Figure 15 and Figure 16A,B). The tetrads of the outermost stratum 1 constitute pollen walls with a three-layered structure (Figure 16C,D). The outer compact sporopollenin wall layer (layer 1) forms a continuous compact wall layer, but only on the distal half of the outer pollen grains of the tetrad (Figure 16C). This layer decreases in thickness towards the contact zones of adjacent tetrads and transitions into sporopollenin granules that extend inwardly into an electron-translucent pollen wall layer (layer 2) subjacent to layer 1 (Figure 16C,D). Layer 2 surrounds each pollen grain (monad) and forms a common layer around each tetrad. As staining with potassium permanganate did not indicate a lipidic nature of layer 2, it is most likely polysaccharidic (Figure 17D,E).The inner layer (layer 3) is most likely a monolayered intine. 

**Figure 9 plants-13-01114-f009:**
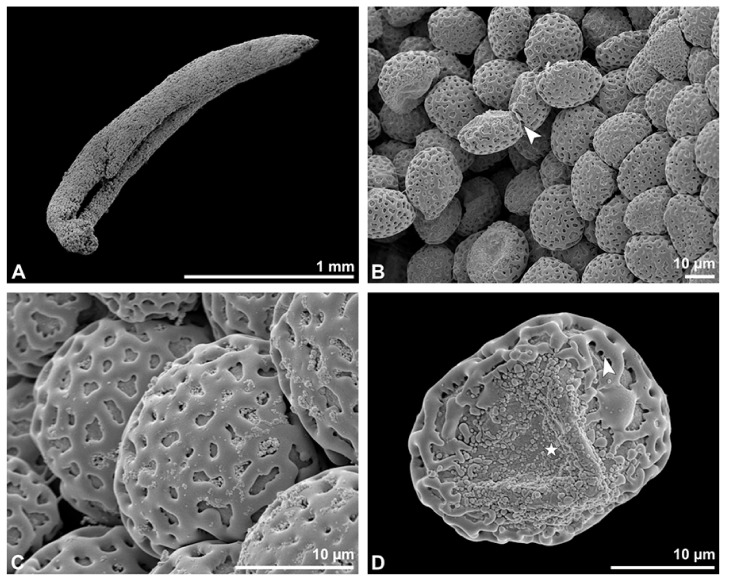
SEM micrographs of hydrated *Cephalanthera longifolia* pollinium/pollen (EtOH + CPD). Note: (**A**) Overview of pollinium. (**B**) Arrangement of monads in the pollinium; remnant of a tetrad (white arrowhead). (**C**) Distal pole with reticulate, heterobrochate ornamentation. (**D**) Oblique polar view of a monad with ornamented an aperture membrane (white star) and a reticulate interapertural area showing a columellate infratectum (white arrowhead).

**Figure 10 plants-13-01114-f010:**
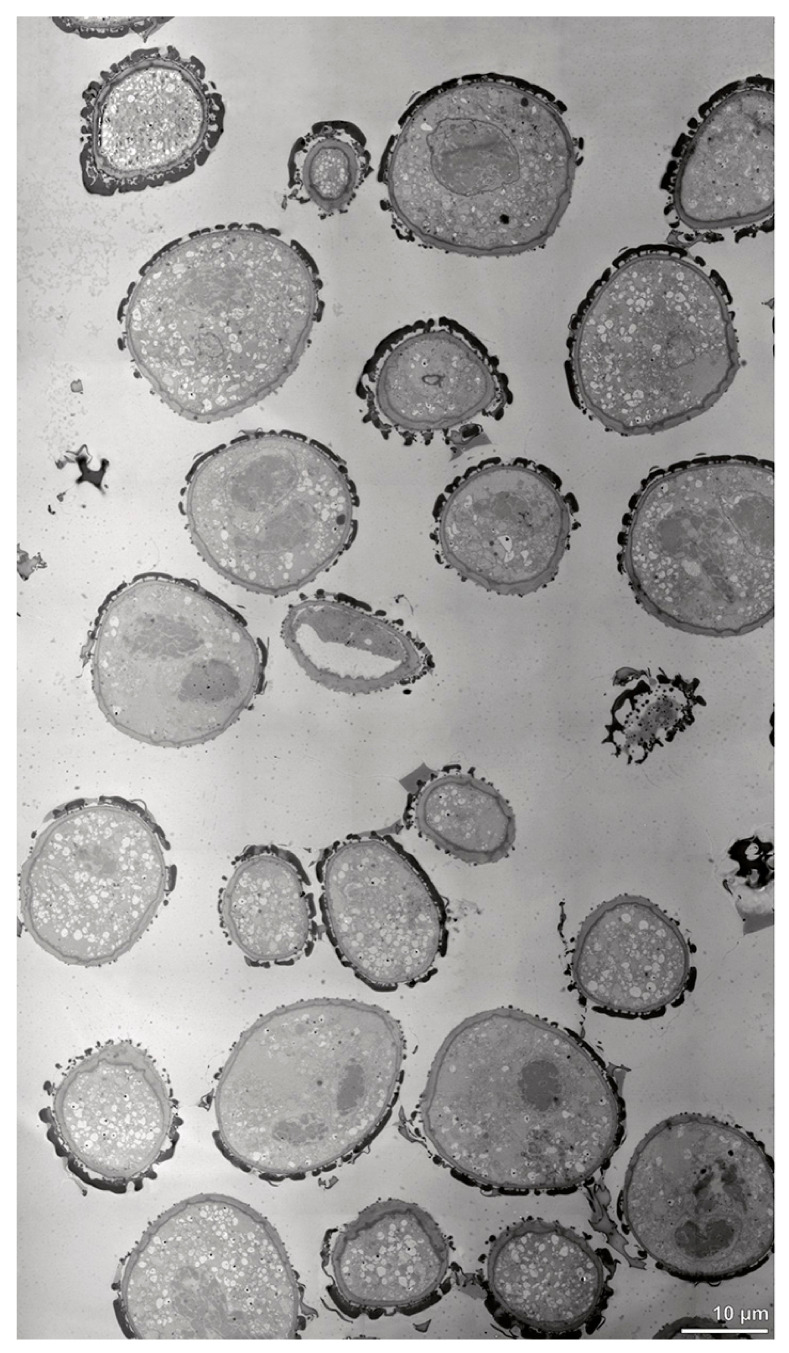
TEM panorama micrograph showing the cross-section of a *Cephalanthera longifolia* pollinium.

**Figure 11 plants-13-01114-f011:**
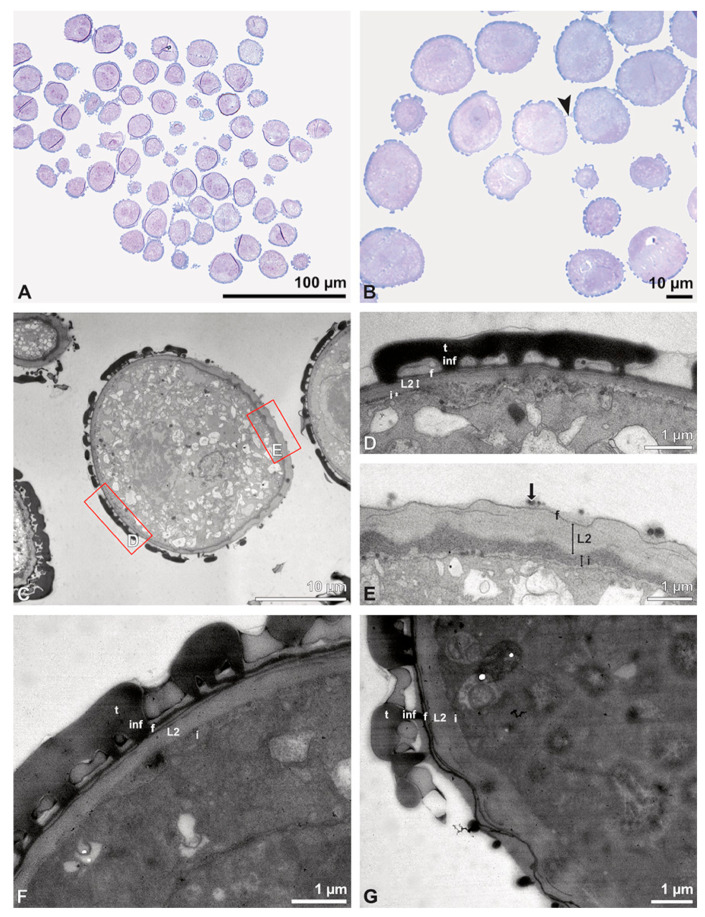
LM (**A**,**B**) and TEM (**C**–**G**) micrographs showing the cross-sections of a *Cephalanthera longifolia* pollinium/monad. Note: (**A**) Semi-thin section of pollinium showing monads, TBO. (**B**) Semi-thin section showing the sticky material (black arrowhead) connecting monads, TBO. (**C**) Overview of a monad. (**D**) Detail of pollen wall in interapertural area, showing semitectate tectum (t), columellate infratectum (inf), continuous foot layer (f), subjacent layer 2 (L2), and a thin intine (i). (**E**) Detail of pollen wall in aperture area; ornamented aperture membrane (black arrow) consisting of intine (i), layer 2 (L2) with lamellated interstratifications (black arrowhead), and a thin foot layer (f). (**F**) Pollen wall in the interapertural area, KMnO_4_. (**G**) Pollen wall in the transition zone between the interapertural and aperture areas, KMnO_4_.

**Figure 12 plants-13-01114-f012:**
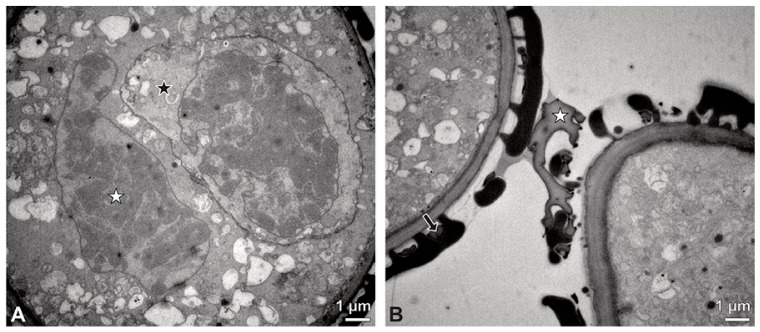
TEM micrographs showing cross-sections of a *Cephalanthera longifolia* pollinium/monad. Note: (**A**) Mature two-celled pollen grain with a vegetative nucleus (white star) and generative cell (black star). (**B**) Monads with two different pollen coatings, electron-dense in the infratectum (black arrow) and between monads (white star).

**Figure 13 plants-13-01114-f013:**
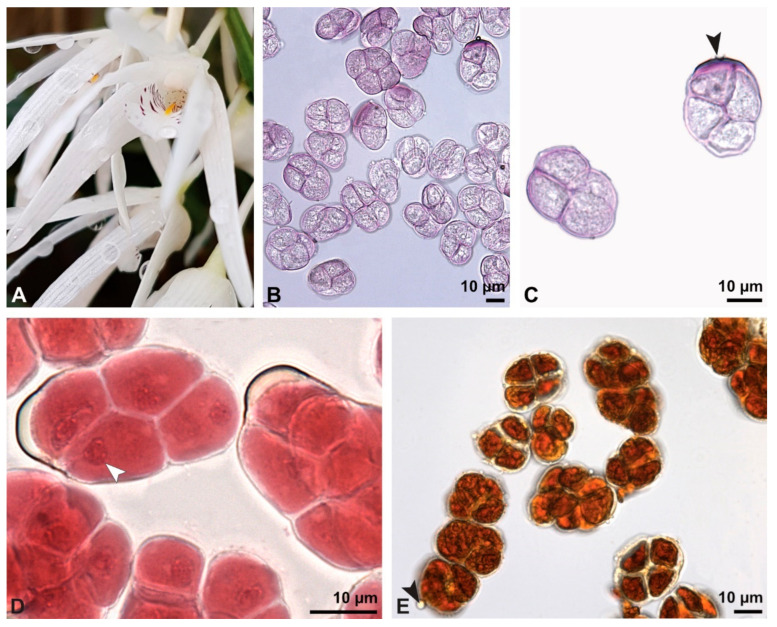
Macroscopic image of flowers (**A**) and LM micrographs of *Dendrobium* × *delicatum* pollen (**B**–**E**). (**B**) Various tetrad types, basic fuchsin. (**C**) Outer tetrad with sporopollenin exine stained dark violet (black arrowhead) and inner thin-walled tetrad, basic fuchsin. (**D**) Binucleate pollen and its generative cell (white arrowhead), Carmine. (**E**) Pollen coatings present (black arrowhead); no starch detected, Lugol.

**Figure 14 plants-13-01114-f014:**
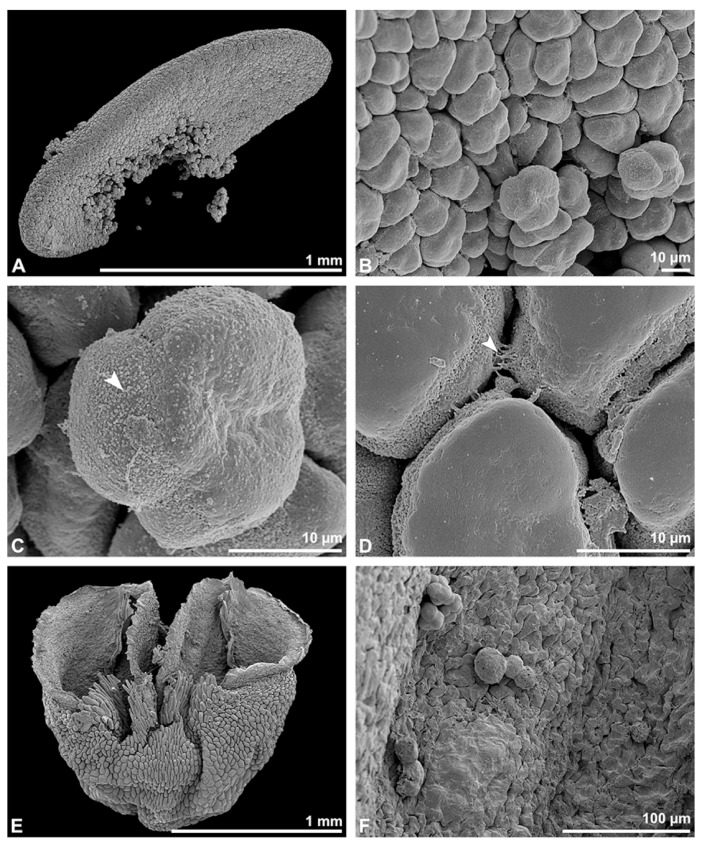
SEM micrographs of *Dendrobium* × *delicatum* pollen. Note: (**A**) Overview of hydrated pollinium (DMP + CPD). (**B**) Arrangement of tetrads within pollinium. (**C**) Tetrahedral tetrad with granulate to microverrucate ornamentation (white arrowhead). (**D**) Outer tetrads with psilate ornamentation and pollen coatings (white arrowhead). (**E**) Anther cap. (**F**) Inner surface of anther cap without Ubisch bodies.

**Figure 15 plants-13-01114-f015:**
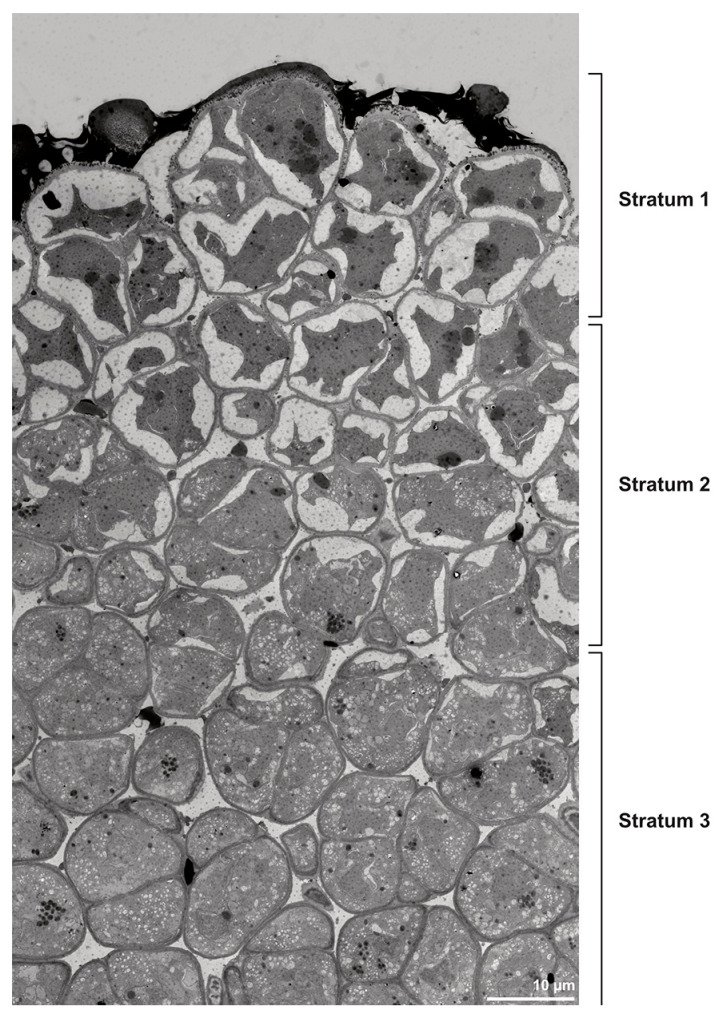
TEM panorama micrograph showing a cross-section of a *Dendrobium* × *delicatum* pollinium. The pollinium can be divided into three strata based on the variations in its tetrad/pollen wall structure and pollen protoplast ultrastructure. This stratification can be traced from the peripheral margin (stratum 1) inward (stratum 2 and 3).

**Figure 16 plants-13-01114-f016:**
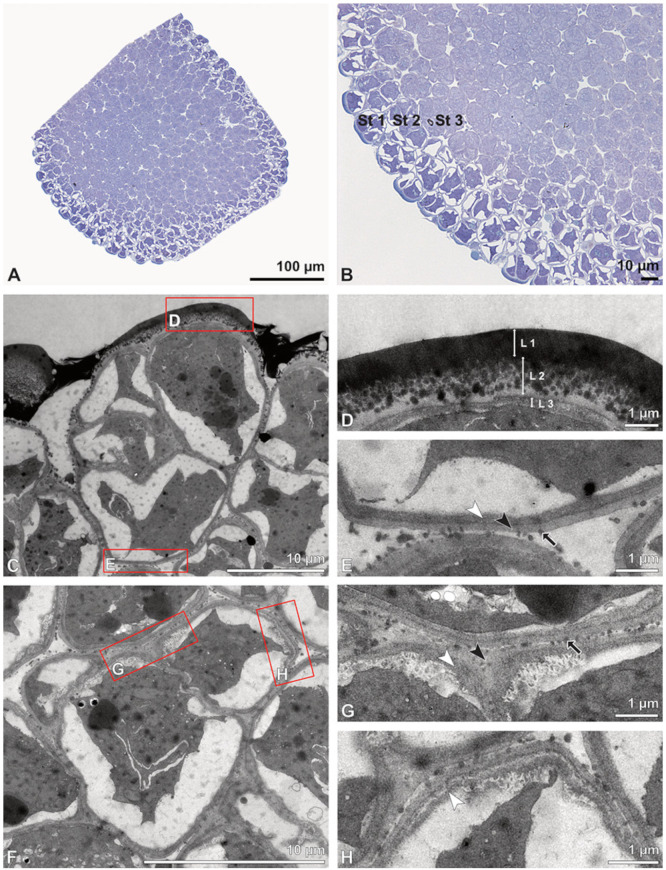
LM (**A**,**B**) and TEM (**C**–**H**) micrographs showing cross-sections of *Dendrobium* × *delicatum* pollinium/tetrads. Note: (**A**,**B**). Semi-thin sections of pollinium, showing different strata (St 1, St 2, St 3), TBO. (**C**). Tetrad of outer stratum 1 with detached protoplasts and three- (**D**) to two-layered (**E**) pollen walls. (**D**) Detail of pollen wall with three layers (L1, L2, L3). (**E**) Detail of pollen wall with layer 2 (black arrowhead) with loosely scattered granules (black arrow) and layer 3 (intine, white arrowhead). (**F**) Tetrad of stratum 2 with detached protoplasts and two-layered pollen wall. (**G**) Detail of pollen wall, layer 2 (L2, black arrowhead) with granules (black arrow) and layer 3 (L3, intine, white arrowhead). (**H**) Detail of pollen wall with distally undulating area of L3 (intine, white arrowhead).

**Figure 17 plants-13-01114-f017:**
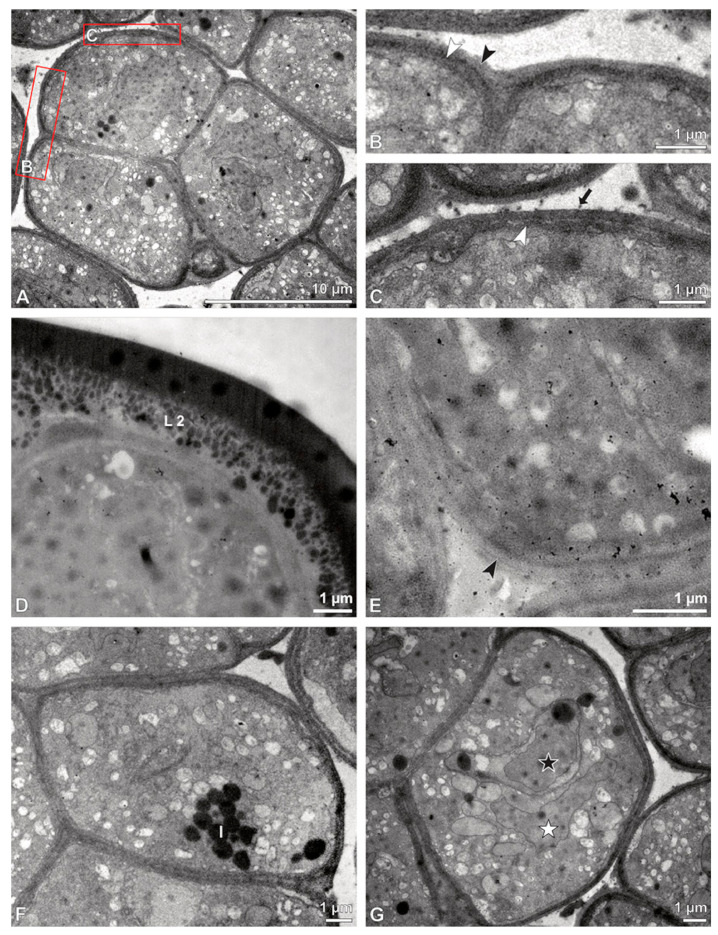
TEM micrographs showing cross-sections of *Dendrobium* × *delicatum* pollinium/tetrads. Note: (**A**) Tetrad of inner stratum 3 with reduced pollen wall. (**B**). Detail of pollen wall with layer 2 (L2, black arrowhead) and layer 3 (L3, intine, white arrowhead). (**C**) Detail of pollen wall with distally undulating layer 3 (intine, white arrowhead) and granules (black arrow) attached to layer 2. (**D**) Pollen wall of tetrad from outer stratum 1, with granules embedded in layer 2 (L2), KMnO_4_. (**E**) Pollen wall of tetrad from inner stratum 3; arrowhead points to L2, KMnO_4_. (**F**) Pollen grain with lipid droplets (l). (**G**) Mature two-celled pollen grain with a generative cell (black star) and vegetative nucleus (white star).

**Figure 18 plants-13-01114-f018:**
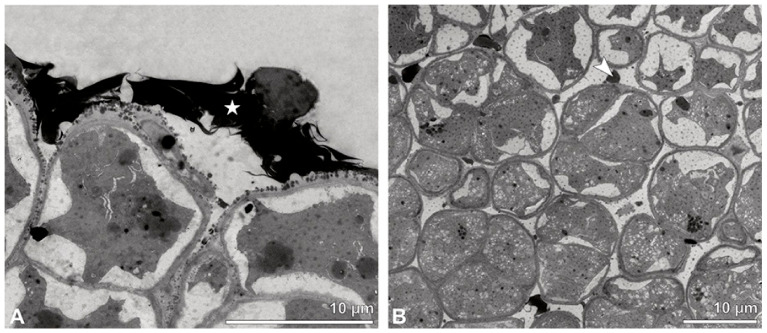
TEM micrographs showing cross-sections of *Dendrobium* × *delicatum* pollinium/tetrads. Note: (**A**) Pollen coatings (white star) attached to the tetrads of the outer stratum 1. (**B**) Pollen coatings (white arrowhead) between the tetrads of stratum 2 (with detached protoplasts) and the inner stratum 3.

The inner pollen grains of the outer tetrads exhibit a reduced pollen wall made of a layer 2 with loosely scattered sporopollenin elements and a monolayered layer 3 (Figure 16E). The pollen grains in the outer tetrads have detached protoplasts lacking intact nuclei and organelles, which indicates that they have no germination function (Figure 15). Stratum 2 consists of tetrads with reduced pollen walls (Figure 16F), comprising only layer 2 and layer 3 (Figure 16G). Wall layer 2 contains loosely scattered embedded sporopollenin granules, surrounding the monads and additionally forming a common layer around the tetrads (Figure 16F,G). The subjacent wall layer 3 passes into a undulated area in the distal half of the pollen grains (Figure 16H). Similar to the outer tetrads of stratum 1, the pollen grains in this region have detached protoplasts and are poor in organelles, indicating that they have no germination function (Figure 15). The tetrads of stratum 3 (Figure 17A) share similarities in their pollen wall structure to those of stratum 2, except that wall layer 2 exhibits very few or no sporopollenin granules (Figure 17B,C). Unlike the tetrads of stratum 1 and 2, the pollen protoplasts are rich in organelles (such as nuclei, vacuoles, lipids) and appear to have a germination function (Figure 17F). Starch reserves have not been detected at this stage of pollen development (Figure 13E). Pollen coatings are primarily attached to the ektexine of the outer tetrads (stratum 1) (Figure 18A), with their quantity decreasing inwardly (Figure 15). Droplets of similar electron density are also observed in the spaces between the tetrads of all three strata, suggesting the presence of pollen coatings (Figure 15 and Figure 18B). Staining with potassium iodine in LM also revealed the presence of pollen coatings (Figure 13E). Ubisch bodies were not found in the anther cap in SEM (Figure 14E,F).

Remarks: Pollen studies of *Dendrobium* are rare. Schill and Pfeiffer [24] investigated 28 species including *Dendrobium* × *delicatum* using SEM and described their ornamentation as predominantly psilate, which is in accordance with our results. The LM studies of *D. kingianum* Bidwill ex. Lindl conducted by Wolter and Schill [29] and the TEM studies of *D. nobile* Lindl. by Zavada [39] revealed a reduction in the pollen wall structure from the peripheral to the inner tetrads of the pollinia. The present study confirms this and furthermore specifies this reduction (in wall thickness and sporopollenin content) in the wall structure from the peripheral tetrads of stratum 1 to the tetrads of the inner strata 2 and 3. The pollen wall of *D.* × *delicatum* deviates from the “typical” angiosperm pollen wall model (see [10]) by having a differently structured intine, comprising both an endintine and ectintine. Additionally, protoplast detachment extends over strata 1 and 2, similar to *Bulbophyllum retusiusculum*.

Description of species *Oncidium crocidipterum* (Rchb. f.) M.W.Chase & N.H.Williams (Figure 19, Figure 20, Figure 21, Figure 22, Figure 23 and Figure 24; Table 1, Table 2 and Table 3), higher epidendroids: The pollinarium is irregularly shaped and consists of two pear-shaped pollinia and an appendage, measuring 3.75 mm in length under LM (Figure 19B and Figure 20A, Table 1). Each pollinium shows a posterior suture running from the base to the apex (Figure 20B). The pollinia, 1.5 mm long and 0.98 mm wide (LM), are attached to a 1.82 mm long (LM) cellular appendage referred to as a stipe (Figure 19B and Figure 20A). Both the pollinia and stipe show autofluorescence under LM (Figure 19C). The frenicular caudicle connects the stipe and pollinium (Figure 20C). Each pollinium is composed of tetrads agglutinated by pollen coatings (Figure 24B). Different types of tetrads have been documented: tetrahedral, planar-tetragonal, planar-rhomboid, and decussate (Figure 19D). In LM, the size range of the hydrated pollen grains within a tetrad is between 12.9 µm and 25.6 µm (Table 1). Mature pollen grains are inaperturate and two-celled at anthesis, and pollen reserves like lipids or starch have not been detected at this stage (Figure 19H,I). In LM, germination was observed on pollen hydrated in water and toluidine blue stain (Figure 19G). Staining with toluidine blue shows the differentiation in the pollen wall structure of tetrads depending on their position within the pollinium (Figure 19 E,F). 

The outermost tetrads exhibit a thick pollen wall in LM (Figure 19E), with an outer wall layer displaying a psilate, perforate ornamentation in SEM (Figure 20D). In contrast to the outer tetrads, the inner tetrads have a thin pollen wall (Figure 19F).

The TEM studies revealed three different strata, based on the differentiation in the pollen’s wall structure: the outermost stratum 1 consists of thick-walled outer tetrads, stratum 2 consists of tetrads with reduced pollen walls, and the innermost stratum 3 comprises tetrads with highly reduced pollen walls (Figure 21 and Figure 22B). The tetrads of the outermost stratum 1 exhibit a three-layered pollen wall structure and show slightly different ektexine thicknesses depending on their position (intersutural-intrasutural) within the pollinium (Figure 22A). The tetrads in the intersutural area (Figure 22C) constitute a compact, continuous sporopollenin layer (layer 1), located only on the distal half of the outer pollen grains of the tetrad (Figure 22D). Towards the contact zones of neighbouring outer tetrads of the pollinium, this compact layer 1 transitions into sporopollenin granules, which decrease inwardly in quantity and density (Figure 22C). The thick, compact electron-translucent layer 2 shows embedded sporopollenin granules with a decreasing density gradient from layer 1 to layer 3 (Figure 22D). This layer is remarkably thicker distally (Figure 22C) and surrounds each monad, also forming a common wall layer around each tetrad (Figure 22C–E). The embedded scattered granules in layer 2 decrease in density inwardly (Figure 22C). These granules are similar in electron density to layer 1 (Figure 22D) and, when stained with potassium permanganate, they become electron-dense in TEM, indicating their lipidic nature (Figure 23E). From a chemical point of view, layer 2 is most likely composed of polysaccharides, since staining with potassium permanganate did not indicate that it has a lipidic nature (Figure 23E,F). The inner pollen wall layer (layer 3) likely constitutes a monolayered intine (Figure 22D,E). Contrary to the outer tetrads in the intersutural area, outer tetrads in the intrasutural area (Figure 22A) display an ektexine composed of a thinner and more electron-translucent distal layer 1 and a subjacent distally thicker layer 2 with only a few embedded sporopollenin granules (Figure 22A,F). Layer 3 is a monolayered intine (Figure 22G). The inner pollen grains of the outer tetrads of both areas, intersutural and intrasutural, have a reduced pollen wall made of a thin layer 2 with scattered embedded granules and a thin monolayered layer 3 (intine) (Figure 22E,H). The pollen grains of stratum 1 show detached protoplasts containing a vegetative nucleus, a generative cell, and many small vacuoles (Figure 22C,F). Stratum 2 consists of tetrads with a reduced pollen wall structure, composed of layer 2 and layer 3 (Figure 23A,B). Layer 2, displaying embedded loosely scattered granules, constitutes the outer pollen wall of each monad and forms a common layer around the tetrads (Figure 23A,B). The inner layer 3 is monolayered (Figure 23B). As with tetrads of stratum 1, the detached protoplasts and ultrastructure of the cells’ content suggest infertility (Figure 23A). The tetrads of stratum 3 display highly reduced pollen walls composed of a layer 2 with only a few sparsely embedded granules and a monolayered layer 3 (Figure 23C,D). Layer 2 surrounds each monad, varies in thickness, and forms a common layer around each tetrad (Figure 23C). The protoplasts of the pollen grains are detached, containing a vegetative nucleus, a generative cell, and many small vacuoles (Figure 23C). An additional thin continuous electron-dense layer surrounds all tetrads of all strata (Figure 22E and Figure 23B–D). Pollen coatings are present primarily in the intrasutural area attached to the outer tetrads and also between the tetrads of stratum 1 (Figure 24A). The spaces between tetrads of all strata are filled with an electron-translucent material, probably elastoviscin (Figure 24B).

**Figure 19 plants-13-01114-f019:**
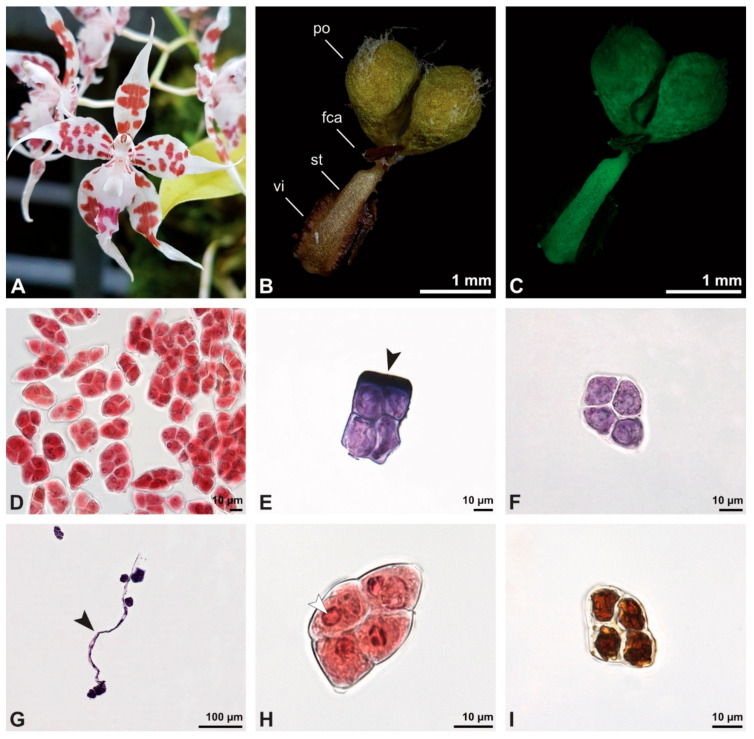
Macroscopic image of flowers (**A**) and LM micrographs of *Oncidium crocidipterum* pollen (**B**–**I**). (**B**) Pollinarium consisting of two pollinia (po), a frenicular caudicle (fca), stipe (st), and viscidium (vi). (**C**) Pollinarium fluorescing under UV light. (**D**) Various tetrad types, Carmine. (**E**) Outer (planar-tetragonal) tetrad with its exine stained dark blue (black arrowhead), TBO. (**F**) Inner tetrad with thin pollen wall, TBO. (**G**) Germinating pollen grain with pollen tube (black arrowhead), TBO. (**H**) Binucleate pollen with a generative cell (white arrowhead), Carmine. (**I**) No starch detected, Lugol.

Remarks: To date, there are no pollen studies on this genus. In contrast to the investigated species above, the PDU type of *Oncidium crocidipterum* constitutes a pollinarium (versus pollinia) with two compact pollinia, a frenicular caudicle, a stipe, and a viscidium. The pollen wall’s structure deviates from the “typical” angiosperm pollen wall model (see [10]) and shows a reduction from the peripheral zones to the centre of the pollinium, as observed in *Bulbophyllum retusiusculum* and *Dendrobium* × *delicatum*. Similar to *D.* × *delicatum*, strata 2 and 3 exhibit only sporopollenin granules that decrease in size and quantity towards the centre of the pollinia. In contrast to *B. retusiusculum* and *D.* × *delicatum*, there is an additional difference in the wall structure of the outermost tetrads from stratum 1 between the inter- and intrasutural area in *O. crocidipterum*. Only in this species does a thin electron-dense layer occur between tetrads, most likely constituting a remnant from the pollen mother cell. Like in *Cephalanthera longifolia*, two different types of sticky materials are present within the pollinium.

**Figure 20 plants-13-01114-f020:**
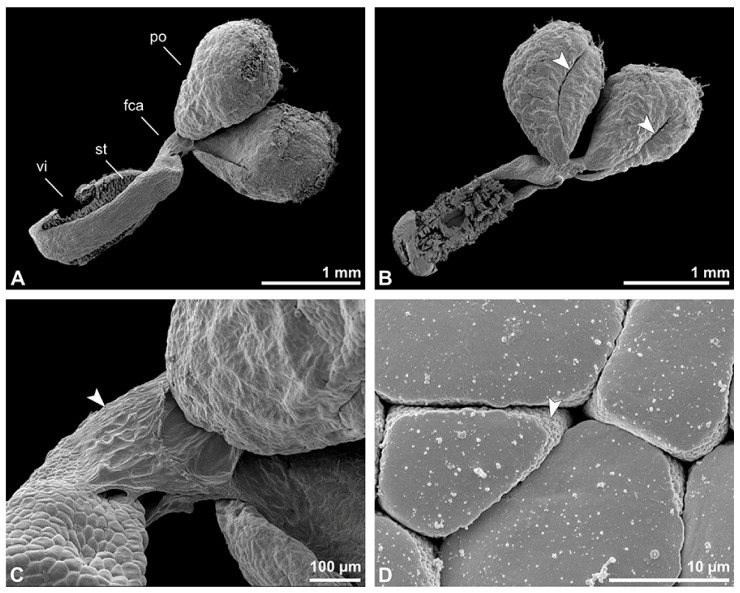
SEM micrographs of *Oncidium crocidipterum* pollinarium/pollen. Note: (**A**) Overview of hydrated (DMP + CPD) pollinarium consisting of two pollinia (po), a frenicular caudicle (fca), stipe (st), and viscidium (vi). (**B**) Dorsal view of pollinarium with two pollinia, each showing a suture (white arrowheads). (**C**) Detail of frenicular caudicle connecting the pollinia and stipe (white arrowhead). (**D**) Exine of outer tetrads with psilate to granulate (arrowhead) ornamentation.

**Figure 21 plants-13-01114-f021:**
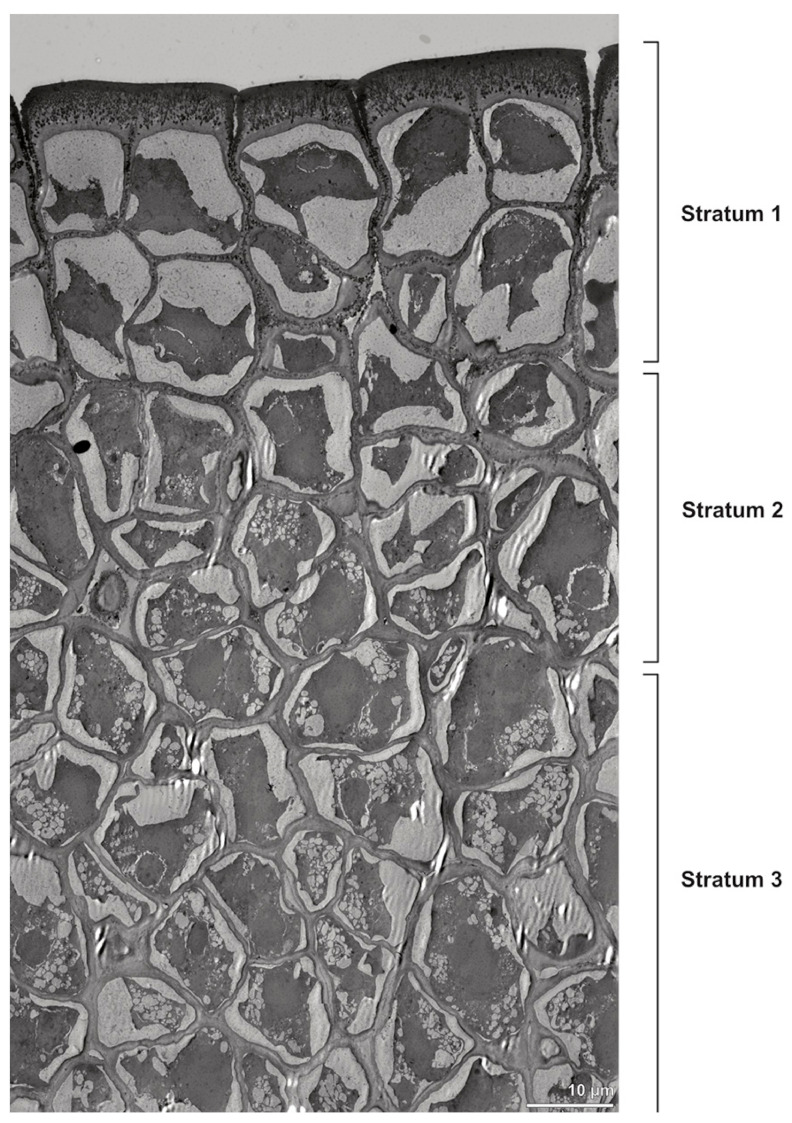
TEM panorama micrograph showing a cross-section of an *Oncidium crocidipterum* pollinium. Note: The pollinium is divided into three strata based on variations in its tetrad/pollen wall structure. Stratum 1 is composed of thick-walled tetrads, stratum 2 of tetrads with reduced pollen walls, and stratum 3 shows tetrads with highly reduced pollen walls.

**Figure 22 plants-13-01114-f022:**
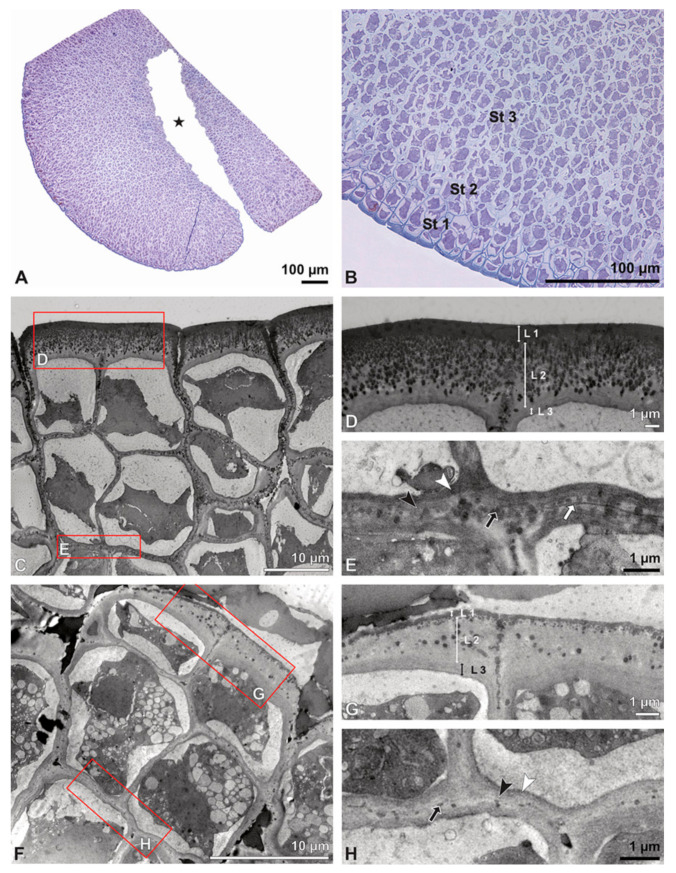
LM (**A**,**B**) and TEM (**C**–**H**) micrographs showing cross-sections of *Oncidium crocidipterum* pollinium/tetrads. Note: (**A**) Semi-thin section of pollinium; black star marks intrasutural area, TBO. (**B**) Semi-thin section showing different strata (St 1, St 2, St 3) based on pollen wall differentiation, TBO. (**C**) Tetrad in intersutural area of outer stratum 1, showing detached protoplasts and a three- (**D**) to two-layered (**E**) pollen wall. (**D**) Detail of pollen wall with three layers (L1, L2, L3). (**E**) Detail of pollen wall consisting of layer 2 (black arrowhead) with embedded granules (black arrow) and layer 3 (intine, white arrowhead), as well as a thin electron-dense layer (white arrow). (**F**) Tetrad in intrasutural area of outer stratum 1, showing detached protoplasts and a three- (**G**) to two-layered (**H**) pollen wall. (**G**) Detail of pollen wall with three layers (L1, L2, L3). (**H**) Detail of pollen wall consisting of layer 2 (black arrowhead) with granules (black arrow) and layer 3 (intine, white arrowhead).

**Figure 23 plants-13-01114-f023:**
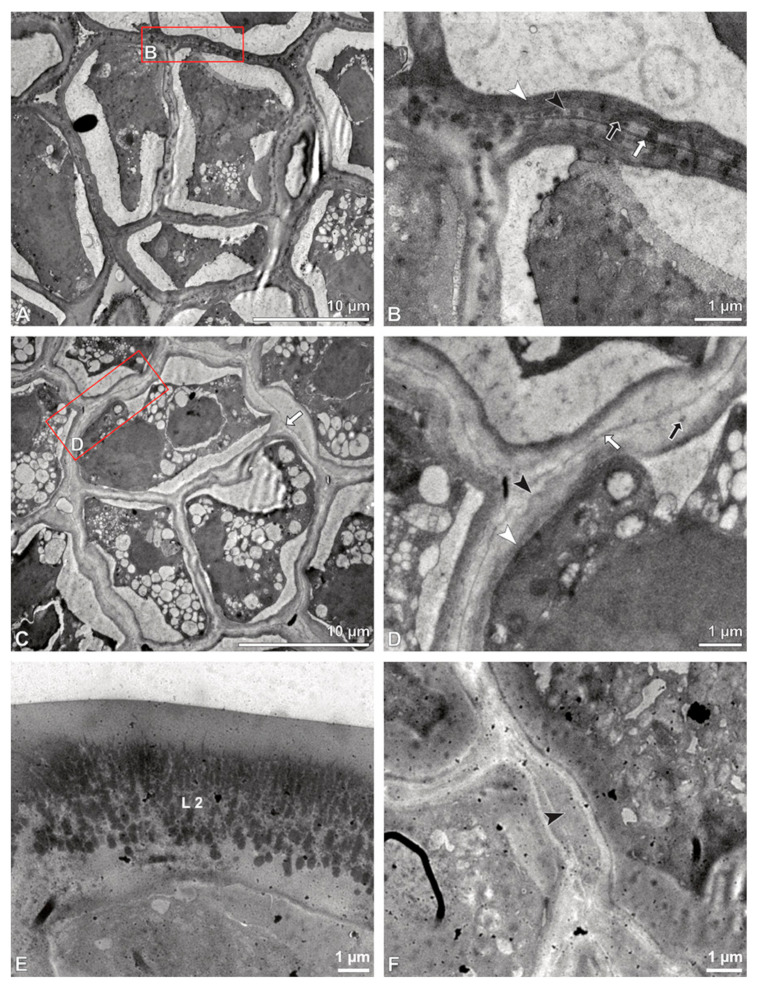
TEM micrographs showing cross-sections of *Oncidium crocidipterum* pollinium/tetrads. Note: (**A**) Tetrad of stratum 2, showing a two-layered pollen wall. (**B**) Detail of pollen wall, showing layer 2 (black arrowhead) with granules (black arrow) and layer 3 (intine, white arrowhead), along with a thin electron-dense layer (white arrow). (**C**) Tetrad of stratum 3, showing a two-layered pollen wall (**D**) and thin electron-dense layer (white arrow). (**D**) Detail of a pollen wall consisting of layer 2 (black arrowhead) with few granules (black arrow), layer 3 (intine, white arrowhead), and a thin electron-dense layer (white arrow). (**E**) Pollen wall of a tetrad from the outer stratum 1, showing an electron-translucent layer 2 (L2) with electron-dense granules, KMnO_4_. (**F**) Outer pollen wall of a tetrad from stratum 3; black arrowhead points to layer 2, KMnO_4_.

**Figure 24 plants-13-01114-f024:**
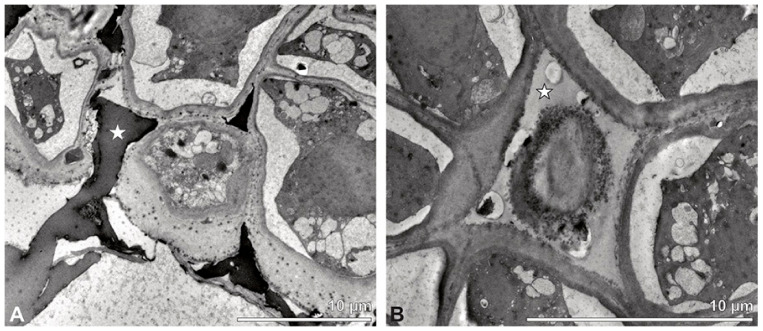
TEM micrographs showing cross-sections of *Oncidium crocidipterum* pollinium/tetrads. Note: (**A**) Pollen coatings (white star) in outermost stratum 1 in intrasutural area. (**B**) Pollen coatings between inner tetrads (white star).

Description of species *Polystachya cultriformis* (Thouars) Lindl. ex Spreng. (Figure 25, Figure 26, Figure 27, Figure 28, Figure 29, Figure 30 and Figure 31; Table 1, Table 2 and Table 3), higher epidendroids: The pollen mass of the whole anther is arranged in four cup-shaped pollinia, with an average length of 0.4 mm and an average height of 0.62 mm under LM (Figure 25B and Figure 26A, Table 1). The pollinia are connected by a sticky material at the base, known as a frenicular caudicle (Figure 27B and Figure 29A), which attaches to a short stipe (Figure 25B and Figure 26D,E). The stipe is 0.33 mm long (LM), exhibits a cellular structure (Figure 26F), and, unlike the pollinium, does not show autofluorescence under LM (Figure 25C). The combination of the pollinia with these appendages is called the pollinarium, which is irregular-shaped and measures 0.92 mm under LM (Figure 26A and Figure 27A). Each pollinium consists of tetrads (Figure 26B) of various types agglutinated by pollen coatings (elastoviscin) (Figure 25I and Figure 26C). The observed tetrad types are planar-tetragonal, planar-rhomboid, tetrahedral, and decussate (Figure 25D). The size of the hydrated pollen grains within a tetrad in LM ranges between 14.8 µm and 18.2 µm (Table 1). At anthesis, mature pollen grains are inaperturate and predominantly two-celled (Figure 25G and Figure 31C). Pollen reserves, like starch, are absent (Figure 25H,I); only some lipid droplets have been detected in TEM (Figure 31C). Toluidine staining of the tetrads in LM revealed differentiation in the pollen wall structure depending on the tetrad’s position within the pollinium, with the outermost tetrads composed of thick wall elements (Figure 25E). 

In SEM, the outer pollen wall layer is psilate (Figure 26C). Compared to the outer tetrads of the pollinium, the inner tetrads have only thin and unstructured pollen walls (Figure 25F).

**Figure 25 plants-13-01114-f025:**
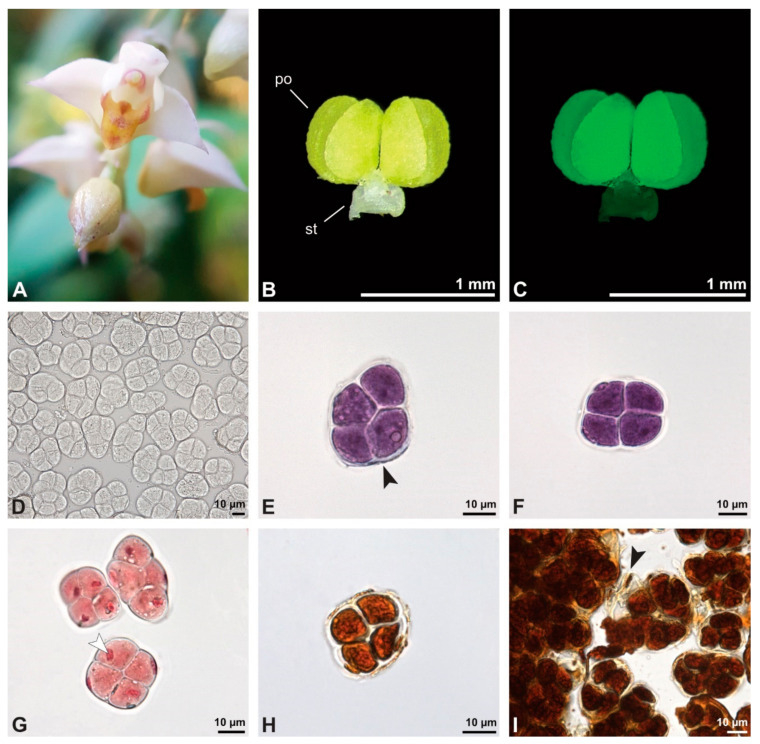
Macroscopic image of flowers (**A**) and LM micrographs of *Polystachya cultriformis* pollen (**B**–**I**). (**B**) Pollinarium consisting of 4 pollinia (po) and a stipe (st). (**C**) Pollinarium excited with UV light. (**D**) Various tetrad types. (**E**) Outer tetrad (planar-rhomboid); exine stained dark blue (black arrowhead), TBO. (**F**) Inner thin-walled (planar-tetragonal) tetrad, TBO. (**G**) Binucleate pollen with a generative cell (white arrowhead), Carmine. (**H**,**I**) No starch detected; pollen coatings present between tetrads (black arrowhead), Lugol.

**Figure 26 plants-13-01114-f026:**
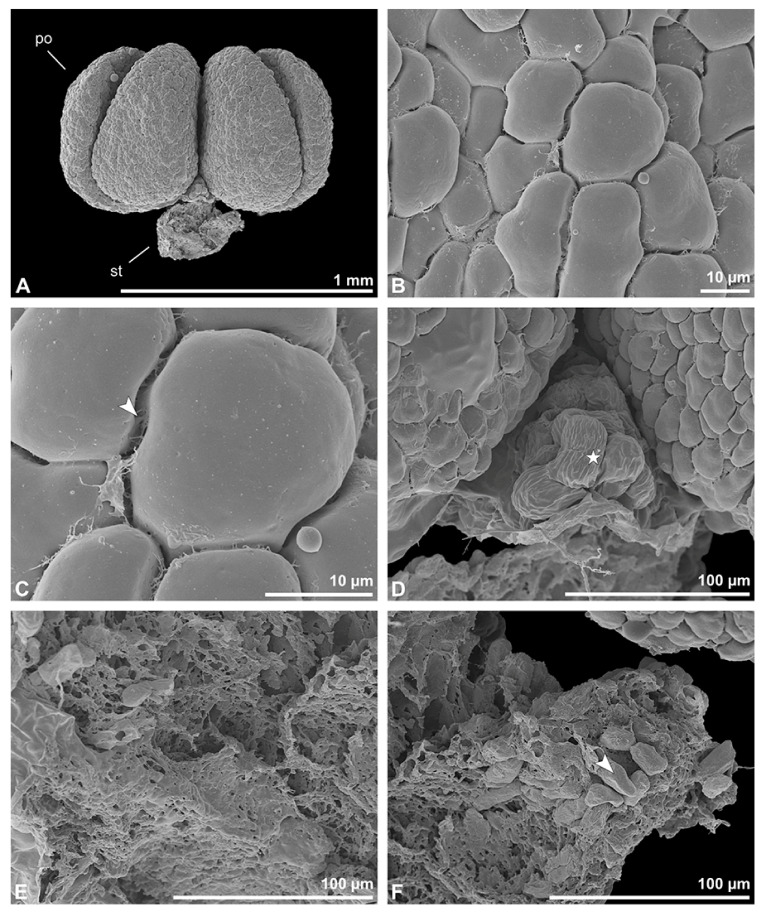
SEM micrographs of *Polystachya cultriformis* pollinarium/pollen. Note: (**A**) Overview of a hydrated (DMP + CPD) pollinarium comprising 4 pollinia (po) and a stipe (st). (**B**) Arrangement of tetrads. (**C**) Tetrads with psilate ornamentation, connected by pollen coatings (white arrowhead). (**D**) Detail of stipe surface (white star). (**E**) Detail of the inner part of the stipe. (**F**) Stipe structure with pollen tetrads attached (white arrowhead).

**Figure 27 plants-13-01114-f027:**
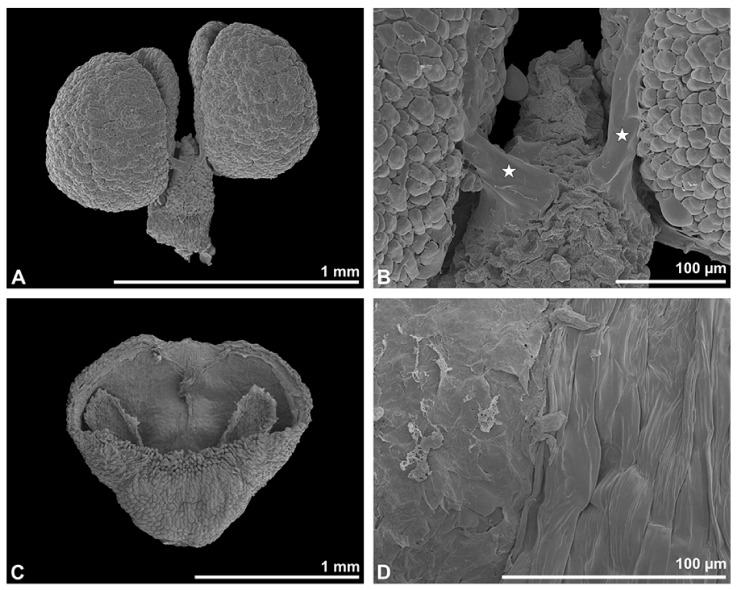
SEM micrographs of *Polystachya cultriformis* pollinarium/pollen. Note: (**A**) Dorsal view of a hydrated (DMP + CPD) pollinarium. (**B**) Frenicular caudicle (white stars). (**C**) Anther cap. (**D**) No Ubisch bodies detected inside the anther cap.

In TEM, this variation in the pollen wall structure of the tetrads is also clearly visible, allowing the pollinium to be subdivided into two distinct strata: stratum 1, with outermost thick-walled tetrads, and stratum 2, with thin-walled inner tetrads (Figure 28 and Figure 29B). The outer tetrads in stratum 1 are characterised by a thick, continuous layer 1 on the distal half of the outer pollen grains (Figure 29C). This layer decreases in thickness inwardly, finally transitioning into granules surrounding the distal half of the inner pollen grains of the outer tetrads (Figure 29C,E). The subjacent electron-translucent pollen wall layer (layer 2) contains electron-dense granules of a consistent distribution only on the distal half of the outer pollen grains (Figure 29D). These granules, exhibiting a contrast similar to that of the ektexine, are thought to consist of sporopollenin. Wall layer 2 surrounds each tetrad, like an ektexine in calymmate tetrads, and also forms the outer wall layer of each pollen grain (monad) (Figure 29C). Staining with potassium permanganate (Figure 30A,B) and the Thiéry test (Figure 30C,D) did not clarify the chemical composition of this layer. Layer 3 constitutes a bilayered intine, composed of an inner electron-dense and an outer electron-translucent polysaccharidic layer (Figure 30C,D). Layer 3 varies in thickness within the tetrad and is slightly thicker in the distal half (Figure 29C). In contrast to the outer tetrads of stratum 1, the inner tetrads of stratum 2 have reduced pollen walls composed of pollen wall layer 2 and layer 3 (Figure 29F). The electron-translucent layer 2 varies in thickness and shows a peripherally diffuse delimitation (Figure 29G). Like in the outer tetrads of stratum 1, this layer surrounds each tetrad and constitutes the outer wall of the pollen grains (Figure 29F). The subjacent bilayered intine (layer 3) is thicker on the distal half of the pollen grains (Figure 29F,G). With the Thiéry test, this layer stains electron-dense, indicating its polysaccharidic nature (Figure 30E,F). The pollen grains are still in a developing stage (mitosis 1), as seen in TEM, where the formation of the generative cell is still visible in most pollen grains (Figure 29F,H and Figure 31A,B). The generative cell/nucleus is formed on the distal half of the pollen grains and is enclosed by an inner, more electron-dense intine layer with an undulating structure (Figure 29F,H). Even within a single tetrad (of both strata), the pollen grains show different developmental stages, ranging from young microspores with nuclei, lipid droplets, and small vacuoles at the middle stage (Figure 31A) to microspores at the late stage (mitosis 1) (Figure 31B), and up to mature two-celled pollen grains with a vegetative nucleus and detached generative cell (Figure 31C). The pollen grains are poor in organelles, with only a few lipid droplets present (Figure 31C). In the pollen grains of the outer tetrads of stratum 1, the protoplast is sometimes slightly detached from the pollen wall (Figure 29C). Pollen coatings are present and attached to the ektexine of the outer tetrads (stratum 1). The coatings differ in their structure, ranging from homogenous to porous (Figure 31D–F). Ubisch bodies on the inner surface of the anther cap have not been detected in SEM (Figure 27C,D).

Remarks: Pollen studies on this genus are sparse. To date, only ultrastructural investigations of *Polystachya concreta* (Jacq.) Garay & H.R.Sweet, by Wolter and Schill [29], and *P. pubescens,* by Schlag and Hesse [40], have been conducted. Both studies describe that the sporopollenin elements are limited to the peripheral zones of the pollinia, which also applies to *P. cultriformis*. While Schlag and Hesse [40] depict a bilayered intine for *P. pubescens*, Wolter and Schill [29] delineate a three-layered intine for *P. concreta.* For *P. cultriformis* we assume a three-layered intine as well. The pollen wall structure of *P. cultriformis* deviates from the “typical” angiosperm pollen wall model (see [10]) and exhibits a reduction in wall structure from the tetrads of stratum 1 (sporopollenin wall elements) to the tetrads of stratum 2 (only granules). Protoplast detachment is infrequent and limited to the tetrads of stratum 1. Furthermore, the sticky materials observed between tetrads and along the peripheral margin of the pollinium differ in structure. 

**Figure 28 plants-13-01114-f028:**
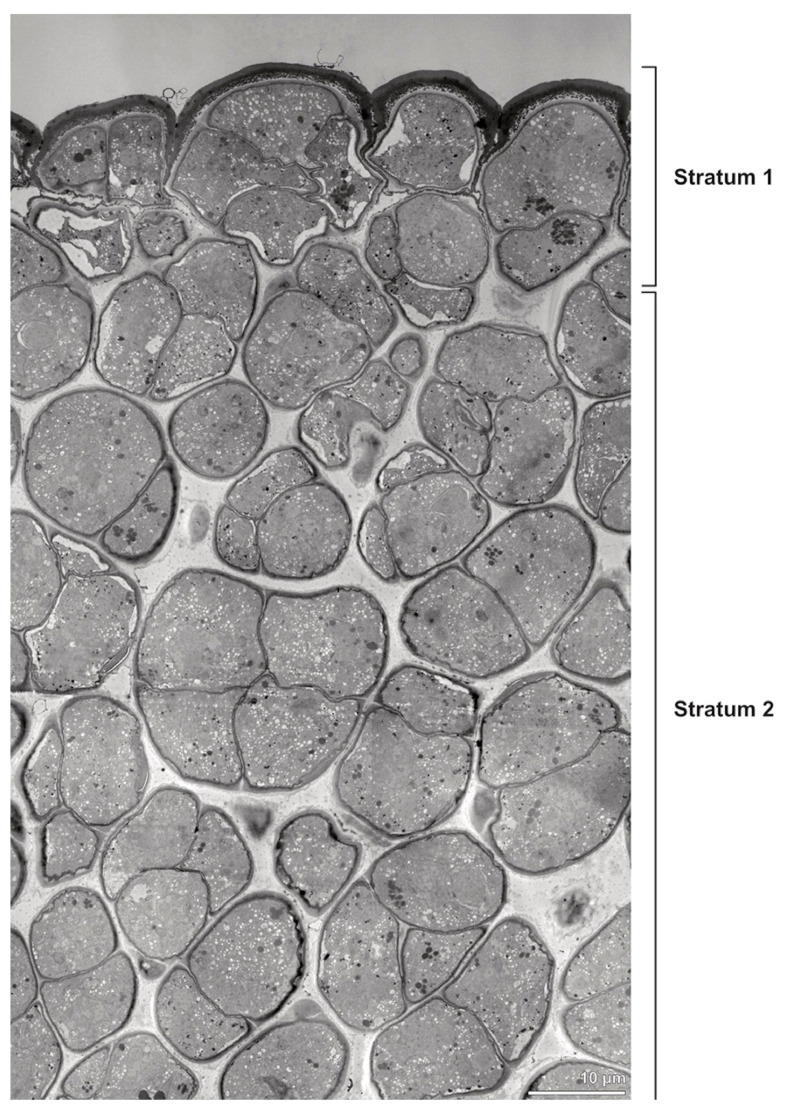
TEM panorama micrograph showing a cross-section of a *Polystachya cultriformis* pollinium. Note: The pollinia are divided into three strata based on the variations in their tetrad/pollen wall structure. The peripheral stratum 1 is composed of thick-walled tetrads and the inner stratum 2 of thin-walled tetrads.

**Figure 29 plants-13-01114-f029:**
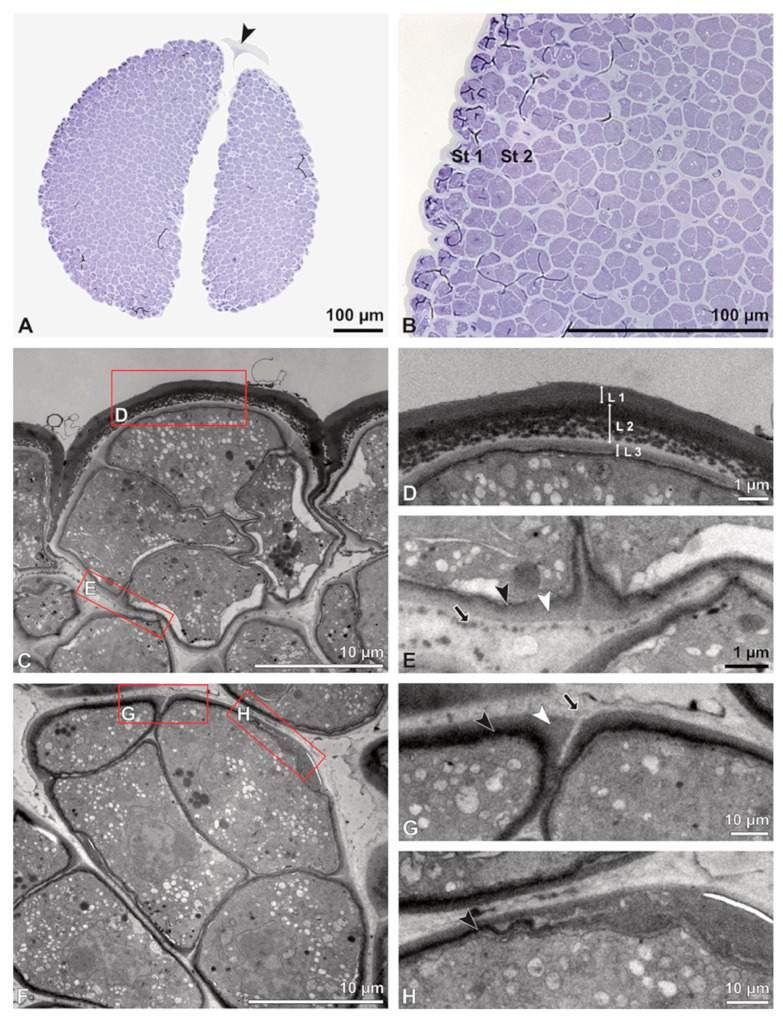
LM (**A**,**B**) and TEM (**C**–**H**) micrographs showing cross-sections of *Polystachya cultriformis* pollinia/tetrads. Note: (**A**) Semi-thin section of two pollinia connected by a sticky material (black arrowhead), TBO. (**B**) Semi-thin section showing different strata (St 1, St 2), TBO. (**C**) Tetrad of outer stratum 1 with slightly detached protoplasts and a three- to two-layered pollen wall. (**D**) Detail of pollen wall with three layers (L1, L2, L3). (**E**) Detail of pollen wall consisting of layer 2 (black arrow), with embedded granules, and layer 3 (bilayered intine, white and black arrowhead). (**F**) Tetrad of stratum 2. (**G**) Detail of pollen wall with layer 2 (black arrow) and innermost layer 3 (bilayered intine, white and black arrowheads). (**H**) Detail of pollen wall layer 3 with inner intine layer (electron-dense) passing into an undulating area on the distal half of the pollen grain (black arrowhead).

**Figure 30 plants-13-01114-f030:**
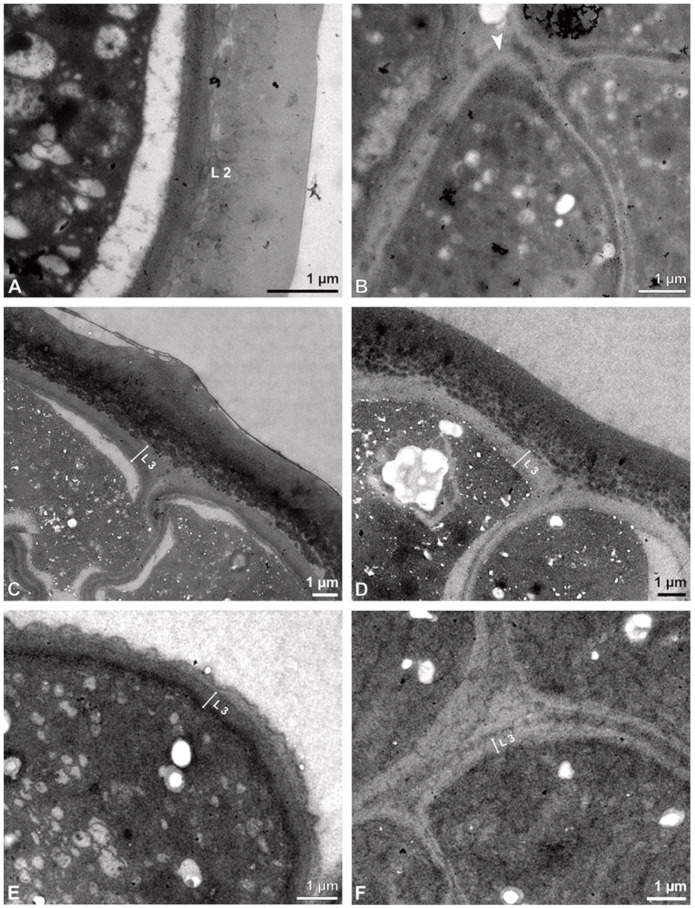
TEM micrographs showing cross-sections of *Polystachya cultriformis* pollinium/tetrads. Note: (**A**) Detail of the pollen wall of a tetrad from the outermost stratum 1, with electron-translucent L2, KMnO_4_. (**B**) Detail of the outer pollen wall of a tetrad from the inner stratum 2; white arrowhead points to L2, KMnO_4_. (**C**–**F**) Electron-dense polysaccharides, Thiéry test. (**C**,**D**) Pollen wall of a tetrad from the outer stratum 1 with an electron-dense intine (L3) and a Thiéry test control (**D**) with an electron-translucent intine (L3). (**E**) Outer pollen wall of a tetrad from stratum 2 with an electron-dense intine (L3) and Thiéry test control (**F**) with an electron-translucent intine (L3).

**Figure 31 plants-13-01114-f031:**
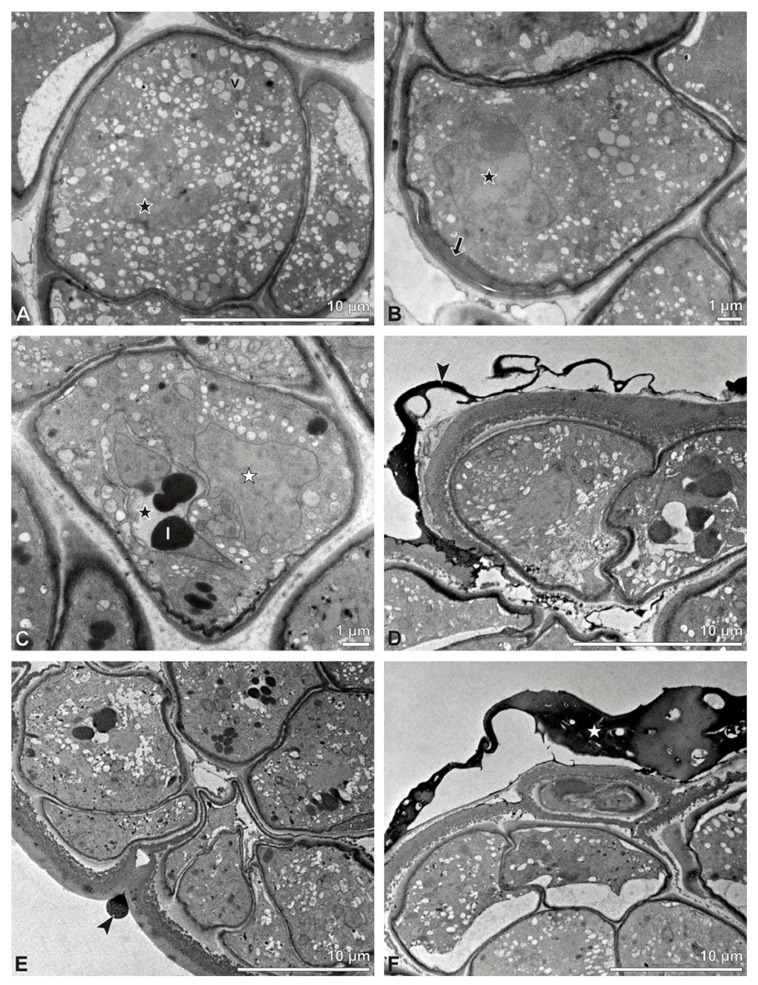
TEM micrographs showing cross-sections of *Polystachya cultriformis* pollinium/tetrads. Note: (**A**) Microspore stage with a vegetative nucleus (black star) and small vacuoles (v). (**B**) Young pollen grain with a vegetative nucleus (black star) close to the pollen wall and the formation of a generative cell on the distal half of the pollen grain (black arrow). (**C**) Mature two-celled pollen grain with a vegetative nucleus (white star), generative cell (black star), and lipid droplets (l). (**D**) Pollen coatings (black arrowhead) attached to the tetrads of outer stratum 1. (**E**) Homogenous pollen coatings (black arrowhead) between adjacent tetrads of outer stratum 1. (**F**) Porous pollen coatings (white star) attached to the ektexine of the tetrads of stratum 1.

## 3. Discussion

### 3.1. Pollen Wall Structure

A “typical” angiosperm pollen wall is composed of an outer exine, subdivided into an ektexine (tectum, infratectum, foot layer) and endexine, and an inner intine [10]. The pollen walls of the investigated epidendroid species deviate from this model, presumably conditioned by their extraordinary agglutination of pollen, leading to their specialised PDU types. Ultrastructural studies have revealed a lack of infratectum, foot layer, and/or endexine, as well as the presence of a unique layer that cannot be compared with any known pollen wall layer, as described by Schlag and Hesse [40] for *Polystachya pubescens* (Lindl.) Rchb.f. Consequently, and contrary to the prevailing literature, for all investigated species except *Cephalanthera longifolia*, more general terms like layer 1, layer 2, and layer 3 were used to describe the specialised ultrastructure of the pollen walls in epidendroids. To relate these layers to literature: layer 1 is referred to as the tectum and layer 3 as the intine. A notable feature among the epidendroid pollinia is the limited distribution of sporopollenin on the outer pollen grains of the outer tetrads of the outer stratum 1, depending on the pollen package type (hard, soft, sectile). In *C. longifolia* (which has a soft package type), every monad displays a sporopollenin ektexine. Among the hard and compact pollinia, which are present in *Bulbophyllum retusiusculum*, *Dendrobium* × *delicatum*, *Oncidium crocidipterum*, and *Polystachya cultriformis,* the sporopollenin layer 1 is limited to the peripheral zones of the pollinium (e.g., [23,28,29]). Based on this limitation of the deposition of sporopollenin to predominantly the outer tetrads among compact, hard pollinia, the exine becomes thicker while saving on “building materials” [29,30]. The inner tetrads of the investigated compact, hard pollinia (*B. retusiusculum, D.* × *delicatum, O. crocidipterum, P. cultriformis*) display reduced pollen walls with sporopollenin elements decreasing towards the centre of the pollinium. Due to the differences in the pollen wall structure resulting from an inward reduction in sporopollenin, the pollinia were subdivided into different strata. A similar differentiation was observed by Schlag and Hesse [40] for *P. pubescens*, where single pollen grains arranged in tetrads were used to distinguish between three different pollen wall types within a pollinium, designated I, II, III. In contrast, the present study considered the tetrads as a unit, and the differentiation into strata was based solely on disparities in the outer walls of the pollen grains within a tetrad, as the internal walls of all tetrads exhibit the same structure. Additionally, the condition of the protoplast (detached or not) was also taken into account when differentiating the strata of pollinia. 

The inhomogeneity of pollen walls primarily arises from the limitation of sporopollenin to the outer tetrads of the pollinia. According to Hesse et al. [30] and Johnson and Edwards [12], this gradual reduction of the exine from the peripheral margin of the pollinium to the centre is conducive to the germination of inner pollen grains. Passarelli and Rolleri [41] suggested that the pollen walls of the inner tetrads are reduced because they are not exposed to dehydration, and, thus, a protective exine is not necessary. Wolter and Schill [29] proposed that the reduction inside the pollinium may be attributed to tightly packed pollen mother cells, sometimes even without intercellular spaces, which prevent the permeation of building materials for wall synthesis within the loculus. However, among the hard, compact pollinia investigated (*D.* × *delicatum, O. crocidipterum, P. cultriformis*), sporopollenin elements, albeit significantly reduced, are also present in the interior, with a mostly decreasing gradient towards the centre of the pollinium. As the sporopollenin precursors and exine formation take place when the microspores are enclosed in callose, the theory of Wolter and Schill [29] might not be appropriate for the early pollen mother cell stage but more fitting for the later stages of the tightly packed pollinium structure. The biosynthesis and localization of sporopollenin have not yet been fully elucidated. Previous studies have assumed that the building materials for sporopollenin synthesis are either produced by the microspores themselves [42] or by both the tapetum and the microspores [43,44]. Studies related to the formation of the exine have demonstrated that sporopollenin can only be synthesised by the tapetum and not by the microspores themselves (e.g., [45,46,47,48,49,50]). Hence, for the Epidendroideae, only a highly reduced or non-sporopollenin exine layer is likely to be found throughout the inner region of the pollinium, since the permeation of compounds produced by the tapetum is greatly reduced due to the tight packaging of the microspores/tetrads, which often even lacks intercellular spaces. Recent studies by Wang et al. [51] on *Arabidopsis thaliana* (L.) Heynh. (Brassicaceae Burnett) and by Tariq et al. [52] on rice anther tapetum have revealed that all sporopollenin biosynthesis proteins, including those responsible for sporopollenin precursor synthesis, are expressed in the tapetal cells and not in the microspores. After the translation of RNA (Ribonucleic acid) into the tapetum, proteins are gradually transported into the locules and further to the microspores [51]. Except for *C. longifolia*, all investigated pollinia exhibit a thick sporopollenin layer in the peripheral areas near the tapetum. In the pollinia formed by septation (see *B. retusiusculum*, *O. crocidipterum*), a sporopollenin exine is present but reduced in the region of the septum, which dissolves during pollinium development (resulting in a suture) and enables sporopollenin biosynthesis proteins to access the free space. In contrast, in the soft pollinia of *C. longifolia,* all monads have a sporopollenin exine layer. Since this pollen package type exhibits the lowest number of pollen grains [15] and a lower density compared to hard, compact pollinia (demonstrated by TEM and semi-thin cross-sections of mature pollinia), the microspores are presumably less tightly packed, providing more space for the permeation of the sporopollenin precursors produced in the tapetal cells. 

The structure of the ektexine (outer exine layer 1) in the outer tetrads of a pollinium varies within the species studied. The majority of the investigated pollinia (*B. retusiusculum*, *D.* × *delicatum*, *O. crocidipterum*, *P. cultriformis*) exhibit a compact, continuous sporopollenin layer 1. An infratectum is lacking in most examined pollinia, except in *C. longifolia* (Table 2). Additionally, depressions are present in the contact zones of the outer tetrads of most pollinia (e.g., *B. retusiusculum*, *O. crocidipterum*), where proteins could potentially be deposited. Adjacent to the ektexine (layer 1), an electron-translucent layer (layer 2) is present. This layer is present in all peripherally and centrally positioned tetrads within the pollinia of all investigated species. Layer 2 often exhibits electron-dense interstratifications (granules and other sporopollenin elements of various sizes), except for in *C. longifolia*. Due to the similar electron density of these interstratifications and the ektexine (layer 1), they are assumed to consist of sporopollenin. The chemical nature of layer 2 is inconsistent in the literature. Schill and Pfeiffer [24] identified this layer as intine since it is not acetolysis-resistant. In TEM studies on the pollen from cypripedioid and epidendroid orchids [30,31], this layer was also considered an ektintine, with the intine described as bilayered for the investigated species, although detailed arguments are missing. Wolter and Schill [29] also concluded that layer 2 is not an endexine but an intine, based on acetolysis experiments on semi-thin sections, and believed that the outer pollen wall of the inner tetrads of a pollinium can be interpreted as an intine both functionally and ontogenetically. In a study by Hesse and Burns-Balogh [53] on different *Habenaria* Willd. species, layer 2 was assumed to be an intine with embedded sporopollenin elements due to its fibrillar structure. This would be a unique feature for orchid pollinia, as sporopollenin elements are usually not known to be embedded/present within the intine. In contrast, Passarelli and Rolleri [41] also examined species of *Habenaria* and proposed the presence of an endexine. Blackman and Yeung [36] imprecisely described this layer as a nexine (lamellated moderately thick layer) but did not differentiate between the endexine and foot layer. According to Halbritter et al. [10], the term nexine is only used for LM, and in TEM, the term nexine corresponds to the foot layer (nexine 1) and endexine (nexine 2). Schlag and Hesse ([40], p. 27) referred to this layer in *P. pubescence* as a “*fibrous intermediate layer which supports the exine, […] and cannot be paralleled with any known sporoderm layer*”. They assumed that this layer has a callosic nature and represents a fibrous component, probably (1-4)-β-glucans, of the callosic wall. Wolter and Schill [22] also described a callosic wall, which is not electron-transparent, as is typical, but exhibits electron-dense fibrillar structures. These assertions can neither be confirmed nor denied since the affiliation of layer 2 (supposedly an exintine) has not been fully clarified. 

Our study reveals that the electron transparency of layer 2 in unstained sections is not indicative of an endexine, which is usually electron-dense due to its lipidic nature [54]. The staining of ultra-thin sections with potassium permanganate excluded the possibility of layer 2 being an endexine, since potassium permanganate would stain the electron-dense endexine, producing a distinct contrast compared to other layers [54]. Determining the chemical nature of layer 2 of *P. cultriformis* remains challenging, as potassium permanganate staining did not indicate that it was an endexine and the results of the Thiéry test are difficult to evaluate due to the large amount of embedded electron-dense granules it contains. Despite the electron density of these granules following the Thiéry test and their electron transparency following potassium permanganate staining, they are assumed to be composed of sporopollenin, as they show the same contrast as the sporopollenin layer 1 in unstained sections. Staining methods sometimes vary in contrast, as evidenced by studies concerning the detection of endexine [54]. As ultrastructural staining experience with respect to orchids is scarce, the results in the unstained sections were given greater emphasis. Acetolysis treatments resulted in a dissolution of the pollinium of *P. cultriformis*, presumably due to a limitation of sporopollenin to the outer pollen wall layer 1 of the outer tetrads of stratum 1. Based on these findings, it is assumed that layer 2 is polysaccharidic in nature with embedded sporopollenin granules. Consequently, *P. cultriformis* has most likely a three-layered intine, which is rare among angiosperms and only known to occur in some species, e.g., Zingiberaceae Martinov (like *Alpinia vittate* W.Bull, *Etlingera elatior* (Jack) R.M.Sm., *Globba winitii* C.H.Wright, *Hedychium gardnerianum* Sheph. ex Ker Gawl. [55], and Apocynaceae Juss. (*Vinca minor* L.) [56], and in a single gymnosperm, *Hesperocyparis arizonica* (Greene) Bartel (Cupressaceae Gray) [57].

The exine of the investigated epidendroid orchids is composed of an ektexine only, which is in accordance with Wolter and Schill [29]. Strikingly, in all species except *C. longifolia* (belonging to the soft package type), the exine transitions into the subjacent layer to a variable extent, which is deviating from the “typical” pollen wall structure of angiosperms. This transition was also observed by Wolter and Schill [29] and Burns-Balogh and Hesse [25] and varied in the amount of transition seen. Burns-Balogh and Hesse [25] suggested that this transition compensates for the absence of other stabilising elements like a foot layer or endexine. The absence of these layers is possibly due to the conservation of energy. Since pollen grains are arranged in a hard, compact union (pollinium), stabilising structures may be superfluous [25,58]. Still, further studies are necessary to clarify the purpose or formation of this uncommon transition area between the ektexine and its subjacent layer. 

The innermost layer 3 most likely represents a polysaccharidic (pecto-cellulosic) intine, present in every peripherally to centrally positioned pollen grain within the pollinia of the studied species. In all investigated orchid species, except for *P. cultriformis*, the intine is very thin, monolayered, undifferentiated, and lacks interstratifications. Contrarily, the intine in “typical” angiosperm pollen usually contains proteins, in variable amounts, that are visible as electron-dense interstratifications often accumulated in the outer ektintine or throughout the monolayered intine [59]. Both the intine and the sporophytic proteins are synthesised in the microspore cytoplasm, in which the proteins are incorporated into during the developing intine [59,60]. Moreover, the innermost pollen wall layer of the investigated epidendroid pollen appears in the early (during pollen mitosis 1) and also late developmental stages (mature pollen) like the primary cell wall [59,61]. The chemical composition of primary cell walls is similar to the intine; both are composed of different polysaccharides (pectins, cellulose, hemicellulose) and proteins [59]. A striking feature of the investigated epidendroid pollen is that the intine is not stained electron-translucent, as is typical for a fully developed intine [10,54], but more electron-dense. Whether the intine is fully differentiated in the investigated species or might further develop during pollen dispersal remains unclear, but the results are in accordance with other TEM pollen studies [24,29,40] where the orchid intine is also thin and undifferentiated.

If layer 2 is considered an intine, all investigated species except for *P. cultriformis* (three-layered intine) exhibit a bilayered intine consisting of an electron-dense endintine and a less-electron-dense ektintine. This contrasts with the orchid literature, as a bilayered intine is often said to display an electron-dense ektintine that keeps the tetrads together (corresponding to layer 2 herein) and an electron-transparent endintine that surrounds each pollen grain (e.g., [12,25,32] and the references therein). This might be another unique feature of orchids with compound dispersal units (pollinia, pollinaria) and requires further TEM studies.

Pollinia, massulae, and tetrads exhibiting a common wall layer, either continuous or discontinuous, are often described as calymmate and acalymmate in the literature (e.g., [11,12,32] and the references therein). According to Halbritter et al. ([10], p. 440), the definition of calymmate is “*dyads, tetrads, and polyads covered by a continuous exine envelope*”. The definition of acalymmate is “dyads, tetrads, and polyads covered by an exine envelope which is discontinuous at the junctions between monads“. In the present study, these terms are not used, as they are not applicable to the investigated epidendroid pollen. In all pollinia examined (except for *C. longifolia*), the inner tetrads lack an exine, since the controversial layer 2 is most likely polysaccharidic and therefore does not correspond to an exine. Peripheral tetrads in the pollinium exhibit an ektexine predominantly on the distal halves of the outer pollen grains, transitioning into granules inwardly without forming a distinct envelope (either continuous or discontinuous). When considering an entire compact pollinium, a surrounding exine is present, but with interruptions in the contact zones of adjacent tetrads. This would represent an acalymmate unit, but the definition, according to Halbritter et al. [10], only refers to dyads, tetrads, and polyads and not to massula and pollinium.

The exine ornamentation in most of the investigated epidendroid pollinia (*B. retusiusculum*, *D.* × *delicatum*, *O. crocidipterum*, *P. cultriformis*) is psilate, or psilate to fine striate, and sometimes perforate, fossulate and/or granulate, while only *C. longifolia* has a foveolate to reticulate-heterobrochate ornamentation (Table 2). The ornamentation of the outer exine is mostly related to the pollination mode of plants. Zoophilous and autogamous plants are known to have highly ornamented pollen, with a thick, structured sporopollenin exine and pollen coatings attached, whereas the pollen of anemophilous plants is less ornamented, its exine is thin and less structured, and pollen coatings are usually absent [10,62]. According to the literature, the pollinators of the investigated orchid species are very diverse and a correlation between pollinator and pollen morphology or exine micromorphology cannot be assumed [26].

The aperture condition of the investigated epidendroid species is inaperturate, except for *C. longifolia,* which produces ulcerate pollen. The inaperturate aperture condition allows for germination anywhere on the pollen surface instead within a limited region, as in the case of an aperture’s presence. Considering the package of pollen grains among Orchidaceae, this functionally omniaperturate condition is presumably beneficial in terms of germination efficiency. Inaperturate pollen grains are common among monocotyledons [63] and are also typical for most of the investigated epidendroid species (e.g., [24,40,41] and the references therein). The ulcerate pollen of *C. longifolia* is also a typical aperture condition within the Orchidaceae. According to Zavada [28], ulcerate pollen occurs in all subfamilies of Orchidaceae, along with other aperture conditions like sulcate, porate, or inaperturate. The exception is Apostasioideae, which has only sulcate pollen. For Epidendroideae, 12 species with ulcerate pollen are described in PalDat [34], including SEM micrographs showing their pollen grains arranged in a tetrad formation with a distally placed ulcus. Barone Lumaga et al. [32] also described, for the pollen of *C. longifolia* and *C. rubra,* a distal ulcus. However, contrary to these findings, our study of *C. longifolia*, conducted using combined LM/SEM/TEM, revealed an ulcus in a proximal position, clearly observed in a tetrad formation. Proximal apertures are rare among angiosperms and have only been observed in pollen from a few species, such as *Tillandsia leiboldiana* Schltdl. (Bromeliaceae Juss.) [64], *Drosera capensis* L. (Droseraceae Salisb.) [65], *Beschorneria yuccoides* K.Koch (Asparagaceae Juss.), and *Asimina triloba* (L.) Dunal (Annonaceae Juss.) [66].

In contrast to other investigated species, the tetrads of *C. longifolia* are not permanent, and all pollen grains within a pollinium, whether from the innermost or outer tetrads, have equally structured pollen walls. Barone Lumaga et al. [32] described a bilayered intine in the aperture area of the pollen of *C. longifolia* and *C. rubra*, whereas the proximal half of these pollen grains is composed of an ektexine, subdivided into a tectum, columellate infratectum, continuous compact foot layer, lamellate endexine, and thin intine. Based on its staining behaviour and similarities in electron density to that of layer 2, this layer in *C. longifolia* is also assumed to be polysaccharidic in nature, with electron-dense interstratifications, rather than an endexine. Unlike the findings of Barone Lumaga et al. [32], the electron-dense lamellated interstratifications in cell wall layer 2 are not indicative of an endexine, as a lamellated endexine typically forms a thick and continuous layer. We assume that these interstratifications consist of sporopollenin, given their similar electron density to the ektexine and resemblance to the granular embeddings observed in the other studied species. Barone Lumaga et al. ([32], p. 514) described the presence of a “pseudovacuole” in pollen grains of *C. longifolia*, stating that it “*anticipates the cytoplasm collapse*”. In the literature, pseudovacuoles are only mentioned in conjunction with bacteria [67] and animal cells [68,69], and not with plant cells or pollen. Pringsheim [67] suggests that pseudovacuoles contain gas and contribute to floatation in liquid environments, while Vanhecke et al. [68] propose a key function for pseudovacuoles in the regulation of intercellular hydrostatic pressure during blebbing (apoptotic blebs are part of apoptosis). However, neither function of pseudovacuoles seems applicable to the pollen of *C. longifolia*, as it does not have a hydrophilous pollination system, for which floating might be advantageous, and all its monads possess a stable wall structure that makes blebbing unlikely. During our study we observed no pseudovacuoles, only large vacuoles, which likely represent an interrupted early developmental stage of the pollen (before pollen mitosis) [10,70]. Although limited data have been published, such defect pollen grains are found frequently in samples (e.g., [24,71]).

### 3.2. Mature Pollen Stage 

Two of the investigated epidendroid species, *B. retusiusculum* and *D.* × *delicatum*, exhibit degenerated tetrads solely at the peripheral zones of the pollinia, while the inner tetrads appear fertile and well fixed. Given the regular arrangement of these degenerated tetrads, a fixation artefact can be excluded. Although data concerning this phenomenon are rare, Burns-Balogh and Funk [58] also noted a similar occurrence of degenerated outer tetrads in *Sarcoglottis* C.Presl, albeit without providing an explanation. While the function of these degenerated tetrads remains uncertain, they may act as a protective cover for the inner fertile pollen grains, analogous to an airbag for pollination. 

A striking feature observed in *P. cultriformis* is the presence of various developmental stages within a single pollinium and even within individual tetrads. This suggests successive tetrad formation, where a callose wall is formed after each meiotic division [64,72]. In angiosperms, two common types of cytokinesis are observed, successive and simultaneous, resulting in different tetrad types. Tetragonal, T-shaped, decussate, Z-shaped, and linear tetrads arise from successive cytokinesis, while simultaneous cytokinesis produces tetrahedral, rhomboidal, tetragonal, and decussate tetrads [64]. Orchid pollen microsporogenesis is often described as simultaneous in the literature (e.g., [2,3,36]). However, our study indicates that both successive and simultaneous microsporogenesis can co-occur in each pollinium, as evidenced by the presence of tetrads of both cytokinesis types.

During pollen development, a generative cell is formed (first pollen mitosis), which subsequently detaches from the pollen wall [10]. The formation site of the generative cell within pollen grains varies among different plant families, occurring at different positions in the microspore [70,73]. The literature on generative cell formation in orchids presents varied findings. Blackman and Yeung ([36], p. 332) mentioned that “*the generative cell is initially located at one end of the microspore delimited by a wall continuous with the microspore wall*”. Studies by Schlag and Hesse [40] and Aybeke [74] revealed a distally located generative cell, whereas Kant [3] published a shift of the microspore nucleus to the proximal position, followed by generative cell formation. The results of the present study point to distally located generative cell formation, since the TEM micrographs of *P. cultriformis* show vegetative cells during their formation and subsequent detachment from the distal half of the pollen grains. After the completion of mitosis I, the two-celled pollen grains of these orchid species show an undulate area only on their distal half, representing a remnant of the generative cell formation area. This undulation is present in almost all orchid species investigated in this study, except for *C. longifolia* and *O. crocidipterum*, supporting the assumption that the generative cell is formed distally in the microspore.

Regarding cytoplasmic reserves, all investigated epidendroid pollen contain variable amounts of lipids, but no starch, consistent with the existing literature (Table 3) [29,33,36].

### 3.3. Elastoviscin/Pollen Coatings

Pollination in angiosperms is often linked to a vector, in most instances represented by insects. For adhesion to the pollinator, and thus efficient pollen transfer, gluing materials, also called pollen coatings, have evolved. Three different types of pollen coatings have been distinguished so far: pollenkitt, tryphine, and elastoviscin, with the latter being exclusive to orchids [75]. Elastoviscin plays an important role in tetrad cohesion and appendage formation, resulting in pollinia or pollinaria as the pollen dispersal units. 

According to Schill and Pfeiffer [24], Wolter and Schill [22], and Wolter et al. [23], elastoviscin is produced in tapetal cells, typically appearing as droplets that fuse after tapetal cell wall dissolution [22,29]. Hesse and Burns-Balogh [53] described the degenerated pollen grains in *Habenaria*, located in the transition zone between the pollinium and caudicle, as well as within the massulae, serving as an additional source of pollen-connecting substances. The cytoplasm of these degenerated pollen grains, which never reach the final mature state, also contributes, as a sticky material, to the viscid material but should not be called elastoviscin as there are ultrastructural differences separating the two. Pacini and Hesse [14] state that remnants of the pollen mother cell walls also act as a glue to cohere tetrads, forming a pollinium. Unfortunately, the results of the present study cannot verify or disprove these theories, as only mature stages of pollinia were investigated. Although some of the investigated pollinia (*B. retusiusculum*, *D.* × *delicatum*, *O. crocidipterum,* and *P. cultriformis*) contain degenerated pollen grains, predominantly in their peripheral zones (stratum 1), no significant difference was observed in the amount of sticky material between the strata of degenerated pollen grains and the strata containing only intact pollen grains. Based on our results, the distribution of pollen coatings between the pollinia of all investigated epidendroid species varies, and no assumption can be made as to whether only the tapetum or other structures within the pollinium are involved in producing the characteristic sticky materials of orchids. Further studies, especially ontogenetic studies, are necessary to clarify the sites of the production of these materials.

Regarding the chemical composition of elastoviscin, very little is known so far. The only available information is that elastoviscin is a polymer which is lipidic in nature, contains unsaturated fatty acids, and is dissolvable in a chloroform–methanol mixture [12,14,23]. Due to the similarities between elastoviscin and pollenkitt with respect to their origin, beginning of synthesis, release, and function, they are considered homologous [22].

As already mentioned, elastoviscin is described as a sticky material restricted to the Orchidaceae, holding tetrads together, forming appendages, and gluing the pollinia/pollinaria to pollinators. Nevertheless, this sticky material is not always called elastoviscin but also lipid material [53], sticky mixture [53], or coating material [24]. Hesse and Burns-Balogh [53] identified three distinct types of sticky materials in orchids: elastoviscin, a sporoderm/lipid mixture, and cohesion strands, with the last two clearly different from elastoviscin. Our results support the occurrence of different types of sticky materials. Differences in electron contrast were observed in the investigated species (e.g., electron-dense substances in *D.* × *delicatum*, less-electron-dense substance in *B. retusiusculum*) and even within a pollinium (e.g., *O. crocidipterum*), as well as in our fluorescence microscopy experiments (the pollinia and viscidium show autofluorescence whereas their appendages do not). It is not evident in the literature whether the term “elastoviscin” represents all forms of sticky materials or only certain ones, which is probably the reason for the different designations. Based on this, more general terms like “pollen coatings” for the connecting material between tetrads/pollen and “sticky materials” for substances located at the base of the massulae, connecting them, were used herein to describe the pollinia/pollinaria. Further research is needed to elucidate the composition and origin of these different sticky materials in orchids to avoid confusion and, thus, to be able to specify the term “elastoviscin” more precisely.

## 4. Materials and Methods

### 4.1. Plant Material

The pollinia/pollinaria (pollen material) of five Epidendroideae species (Table 4) were collected from plants growing in the Vienna floodplain area of Lobau (Donau-Auen National Park) (*C. longifolia*) and from cultivated plants in greenhouses of the Botanical Garden of the University of Vienna (*B. retusiusculum*, *D.* × *delicatum*, *O. crocidipterum*, *P. cultriformis*). The pollinia/pollinaria were collected at anthesis and investigated using LM, SEM, and TEM. Due to the status of Orchidaceae species as threatened within the Donau-Auen National Park, only a limited amount of pollen material (3 pollinia) was collected, and an herbarium voucher was therefore obtained for *C. longifolia*. The geocoordinates of the plant are listed in Table 4. The specimen blocks with embedded pollen material produced for TEM are stored in the collection of the Department of Botany and Biodiversity Research, University of Vienna, Austria. The investigated orchid species growing in the Lobau area of the Donau-Auen National Park were determined according to Fischer et al. [76]. The plant taxonomy used follows Li et al. [9], POWO [38], and IPNI [77].

### 4.2. Sample Preparations and Macro/Micrograph Documentation

Initial investigations using different light microscopic techniques were performed on all species regarding their autofluorescence, dispersal units, size of pollinarium/pollinium/pollen, cellular condition, presence of starch, and ornamentation. Therefore, pollinia were first rehydrated in water or various other aqueous solutions, depending on the detection method. All processed samples were investigated using an Olympus BX50 and photographed with an integrated Olympus UC90.

#### 4.2.1. White and Fluorescence Light Microscopy—General Overview of PDUs

Whole pollinia/pollinaria were placed on a glass slide and investigated using an Olympus SZX16 (with SDF PLAPO 1XPF objective) equipped with a CoolLED’s pE-300 white excitation light. Micrographs were taken with an Olympus EP50 digital camera. The autofluorescence ability of different pollinarium parts was tested using a filter cube for blue excitation wavelengths and green fluorescence (ET470/40; designed for GFP).

#### 4.2.2. Toluidine Blue Staining (TBO)—General Overview of Pollen

Since pollinia consist of many tetrads compactly glued together by pollen coatings/elastoviscin, the pollinia had to be dissected first. In the case of hard and soft pollinia, a small piece was cut off using either a razor blade or a scalpel; in the case of sectile pollinia, a massula was removed with a tweezer. The dissected piece of pollinium/massula was placed in a drop of toluidine blue until the outer pollen grains were stained dark blue [78]. Subsequently, the sample was transferred to a microscope slide with a recess for better handling during the following processes. One drop of petroleum ether and two drops of chloroform were added to dissolve the pollen coatings/elastoviscin. After evaporation of the chloroform (CHCl_3_), the sample was placed in a drop of water on a standard microscope slide and covered with a coverslip. To study single tetrads and avoid overlays, the piece of pollinium/massula was squeezed carefully with the blunt end of a teasing needle. If the staining was insufficient, a further drop of toluidine blue was added at the edge of the coverslip and permeated by placing a filter paper at the opposite edge of the coverslip.

#### 4.2.3. Basic Fuchsin Staining (Basic Fuchsin)—General Overview of Pollen 

To highlight morphological differences and facilitate the differentiation between pollen wall layers using LM, basic fuchsin staining was applied [78]. A tiny piece of the pollinium of *D.* × *delicatum* was cut with a razor blade, transferred into a drop of basic fuchsin staining solution on a glass slide, sealed with a cover slip, and squeezed with the blunt end of a teasing needle.

#### 4.2.4. Acetocarmine Staining (Carmine)—Detection of Cellular Conditions

Pollinia were initially dissected using a razor blade, scalpel, or tweezer. Following rehydration in a drop of tap water for up to one minute, depending on the sample condition, the pollinium piece/massula was transferred into a drop of 25% or 50% acetic acid (CH_3_COOH) (depending on the amount of pollen coating/elastoviscin in the sample) for up to one minute to reduce the stickiness of the coatings. Subsequently, the sample was placed in a drop of acetocarmine [78] on a glass slide and put on a hot plate for a few minutes until the outer pollen grains were stained dark red. The pollinium piece/massula was then removed from the acetocarmine solution using a micromanipulator, transferred into a drop of glycerine or water on a new glass slide and sealed with a coverslip. To remove tetrads from the compact formation and thus avoid overlays, squeezing with the blunt end of a teasing needle was required.

#### 4.2.5. Potassium Iodine Staining (Lugol)—Detection of Starch

Samples were first dissected as described for toluidine blue staining and rehydrated in a drop of water. A piece of the pollinia was then transferred into a drop of water on a new microscope slide. After adding a drop of potassium iodine [78], it was covered with a coverslip and squeezed.

#### 4.2.6. Semi-Thin Sections—Sections of Complete Pollinia 

The semi-thin sectioning [10] (thickness: 150 to 200 nm; cutting speed: 2 mm/s) of embedded samples was performed with an ultramicrotome (Leica EM UC6), using a histo diamond knife (DIATOME). Sections were stretched with xylene, removed with a loop from the water surface of the knife boat, and transferred into a drop of water on a glass slide. The glass slide was put on a hot plate (70–90 °C) to accelerate the drying process. After the evaporation of water, a drop of toluidine blue was added. The glass slide was again placed on a hot plate for about three to five seconds before the drop of toluidine blue was removed by tilting the glass slide. To ensure the thorough removal of toluidine blue, sections were washed with water. Therefore, a drop of water was pipetted directly onto the sections and removed with filter paper. This process was repeated until the water was no longer stained. In the last step, water was pipetted again onto the sections, and they were carefully detached from the glass slide. A needle or a micromanipulator was used if the sections did not peel off by themselves. Re-drying was necessary to eliminate wrinkles. Therefore, floating sections on the glass slide were again placed on a hot plate to stretch the sections on the slide during evaporation. A drop of glycerine was added to the dried sections, sealed with a coverslip, and documented via LM.

#### 4.2.7. Size Measurements

Size measurements were taken from the LM micrographs (except for *Dendrobium* × *delicatum*) of hydrated pollinaria/pollinia/pollen. In the case of pollinia, the length and width were measured on a single pollinium only (Figure 32A). In the case of pollinaria, the total length of the pollinarium, the length and width of the pollinium/pollinia, and the lengths of its appendices (caudicle, stipe, viscidium) were measured (Figure 32B). Pollen grains, arranged in planar-tetragonal tetrads (dominant type), were measured in a hydrated condition under LM. The measurements relate to the longest/shortest polar axis as well as to the longest/shortest equatorial diameter in the optical cross-section (Figure 32C). For each Epidendroideae species, the size of 12 pollen grains was measured (Table 1).

**Figure 32 plants-13-01114-f032:**
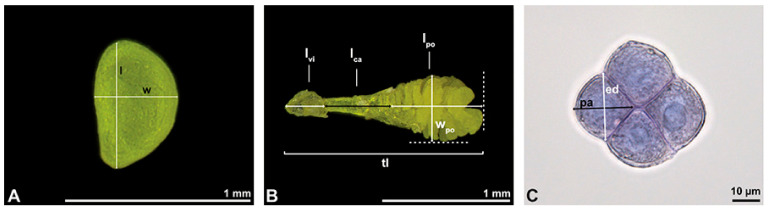
Examples showing measurements of orchid pollinaria/pollinia/pollen grains in their hydrated state using a binocular and compound light microscope. (**A**) A typical compact pollinium; l = length, w = width; binocular micrograph. (**B**) Typical pollinarium; lvi = length of viscidium, lca = length of caudicle, lpo = length of pollinium, wpo = width of pollinium, tl = total length of pollinarium; binocular micrograph. (**C**) Optical cross-section of a typical hydrated tetrad; pa = polar axis, ed = equatorial diameter of an individual pollen grain, LM.

#### 4.2.8. Scanning Electron Microscopy—Ornamentation

The sample preparation for SEM included hydration in water and dehydration with either 2,2-dimethoxypropane (DMP) [79] or a series of ethanol (EtOH, 70–85–96%) and critical point dryings. The pollinia/pollinaria of *Bulbophyllum retusiusculum*, *Dendrobium* × *delicatum*, *Oncidium crocidipterum*, and *Polystachya cultriformis* were dehydrated according to the DMP Direct Method. The pollinia/pollinaria of *Cephalanthera longifolia* were dehydrated using a series of ethanol (EtOH, 70–85–96%). All dehydrated samples were transferred into 100% acetone for 10 to 15 min. Critical point drying was performed with an “Autosamdri^®^-815—Series A tousimis Critical Point Dryer”. Dried pollinia/pollinaria/anther caps were mounted on aluminium stubs and sputter coated with gold in a “BAL-TEC SCD 050 Sample Sputter Coater”. All samples were investigated using a JEOL JSM-IT300 scanning electron microscope.

#### 4.2.9. Transmission Electron Microscopy—Ultrastructures and Chemical Compositions

Fresh samples were chemically fixed either immediately after sampling (*Bulbophyllum retusiusculum*, *Dendrobium* × *delicatum*, *Oncidium crocidipterum*, *Polystachya cultriformis*) or the next day (*Cephalanthera longifolia*). For primary fixation, samples were rehydrated with deionised water for a few seconds before adding phosphate-buffered glutaraldehyde (3%). Postfixation was performed with 1% osmium tetroxide (OsO_4_) and 0.8% potassium ferrocyanide (K_4_[Fe(CN)_6_]) at a rate of 2:1 for about 15 h at 6 °C. After dehydration of the samples with either DMP (*B. retusiusculum*, *D.* × *delicatum*, *O. crocidipterum*, *P. cultriformis*) or an ethanol series (EtOH, 70–85–96%) (*C. longifolia*), they were infiltrated with Agar low-viscosity resin. The fixed and black-stained samples were then cut and transferred into embedding forms filled with freshly mixed Agar low-viscosity resin. The polymerized specimen blocks were cut on a LEICA ultramicrotome (LEICA EM UC6). Semi-thin sections (200 nm) were cut with glass knives (2 mm/s cutting speed) or a DiATOME histo diamond knife (1 mm/s cutting speed). Ultra-thin sections (60–90 nm, 1 mm/s cutting speed) were cut with a DiATOME Ultra 45° diamond knife (standard boat, 3 mm cutting edge). Depending on the staining method, either copper slit-grids or 50 mesh gold grids were used. To verify the presence of the endexine, copper grids with ultra-thin sections were stained in a drop of 1% aqueous Potassium Permanganate (KMnO_4_) solution [54]. The polysaccharides in osmium-free sections were detected according to the Thiéry test [80]. Sections on gold grids were treated with 1% periodic acid (PA) for one hour, 0.2% thiocarbohydrazide (TCH) for 19 h, 7% acetic acid for three minutes, and 1% silver proteinate (SP) for 30 min. For the control, the TCH staining step was skipped. Unsaturated lipids were localized according to the treatment of Rowley and Dahl [81] and Weber [82]. The ultra-thin sections on gold grids were treated with 0.2% thiocarbohydrazide (TCH) for 19 h, 7% acetic acid for 3 min, and 1% silver proteinate (SP) for 30 min. All samples were examined using a Zeiss EM900N Transmission Electron Microscope at 80 kV, with an integrated digital camera (CCD controller), and documented with an Image SP-Program (ISPViewer64). Panorama scans were made at a magnification of 7.000 for better overviews of the pollinia. Picture alignments and picture merges were made using the Image SP-Program (ISPViewer64).

## 5. Conclusions

This study provides new morphological, ultrastructural, and chemical insights into the pollinia/pollinaria/tetrads/pollen of the Epidendroideae of the Orchidaceae family, based on combined LM, SEM, and TEM. The pollen aggregation evident in epidendroid orchids leads to higher morphological diversity and a reduction from thick tectate pollen walls in the peripheral zones of the pollinia to highly reduced pollen walls towards the centre of the pollinium. We can confirm that the PDUs (pollinarium) are more complex in the higher epidendroids, as they have a stipe and frenicular caudicle, while the lower epidendroids only have pollinia. Furthermore, we provide evidence that the middle layer (layer 2) in epidendroid pollen, which has often been declared an endexine, is more likely a polysaccharide layer and should therefore be interpreted as a second or even third intine layer. An additional unique feature unveiled in this study is the aperture condition of *Cephalanthera longifolia*, which exhibits an ulcus in a proximal instead of distal position, as common for this genus. This raises the question of whether this proximal aperture condition is more common among orchids in general or whether *C. longifolia* is an exception. Further studies on pollen from other subfamilies are needed to clarify this. Moreover, our ultrastructural investigations and staining methods for elastoviscin uncovered differences in its contrast, fluorescence ability, and structure, suggesting that there are additional variations in its chemical composition. To elucidate the precise origin of these sticky materials and to refine the term elastoviscin, further chemical and morphological studies are required. In summary, to decipher the complexity of orchid pollen, the combined use of light and electron microscopy is necessary. Moreover, the application of ultrastructural staining methods is crucial for elucidating the chemical nature of pollen coatings, and particularly the orchid feature elastoviscin and the various layers constituting the pollen wall.

## Figures and Tables

**Figure 1 plants-13-01114-f001:**
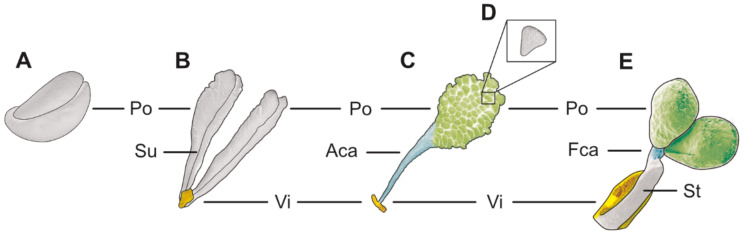
Schematic illustration of different pollinia/pollinaria types. (**A**) Hard, compact pollinia. (**B**) Pollinarium composed of viscidium (vi) and soft pollinia (po) with a suture (su). (**C**) Pollinarium composed of viscidium (vi), appendicular caudicle (aca), and sectile pollinium (po). (**D**) Massula. (**E**) Pollinarium composed of viscidium (vi), stipe (st), frenicular caudicle (fca), and hard, compact pollinia (po).

**Figure 2 plants-13-01114-f002:**
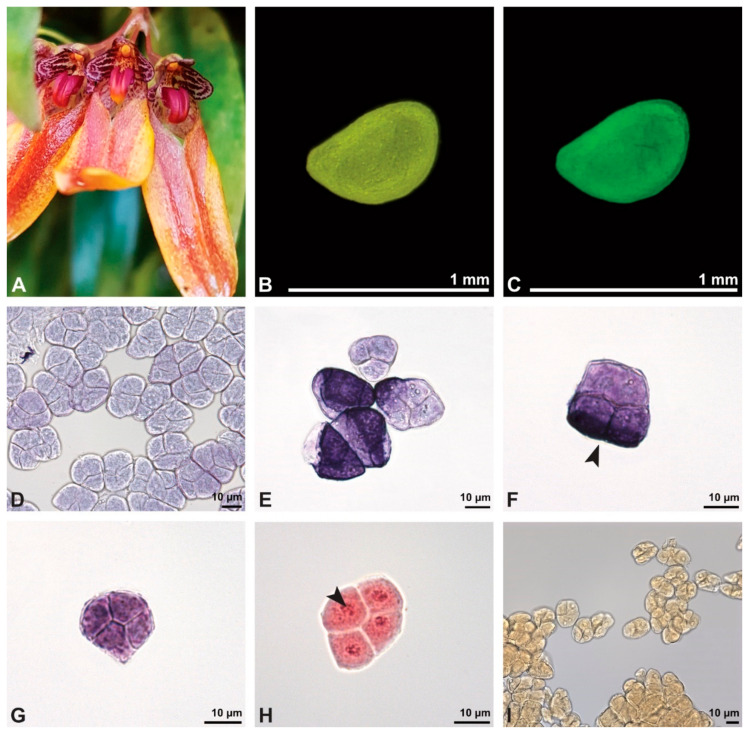
Macroscopic image of flowers (**A**) and LM micrographs of *Bulbophyllum retusiusculum* pollen (**B**–**I**). (**B**) Pollinium. (**C**) Pollinium fluorescing under UV light. (**D**) Various tetrad types, TBO. (**E**) Inner and outer tetrads, TBO. (**F**) Outer (decussate) tetrad with sporopollenin exine stained dark blue (black arrowhead), TBO. (**G**) Inner tetrad with thin pollen wall, TBO. (**H**) Binucleate pollen, generative cell (black arrowhead); Carmine. (**I**) No starch detected, Lugol.

**Table 1 plants-13-01114-t001:** Measurements of pollen grains, accessory structures, and total length of the PDU. Equatorial diameter represents the longest axis of hydrated pollen.

Species	Pollinium Length (mm)	Pollinium Width (mm)	Caudicle/Stipe Length (mm)	Viscidium Length (mm)	Total Length (mm)	Equatorial Diameter (µm)
*Bulbophyllum retusiusculum*	0.69	0.46	-	-	-	10.8–14.3
*Cephalanthera longifolia*	3.2	0.3	-	-	-	26.4–33.1
*Dendrobium* × *delicatum*	1.5 (SEM)	0.5 (SEM)	-	-	-	12.6–15.9
*Oncidium crocidipterum*	1.5	0.98	1.82	-	3.57	12.9–25.6
*Polystachya cultriformis*	0.62	0.4	0.33	-	0.92	14.8–18.2

**Table 2 plants-13-01114-t002:** Morphological and ultrastructural characteristics of pollen from the investigated orchid species.

Species		*Bulbophyllum retusiusculum*	*Cephalanthera longifolia*	*Dendrobium* × *delicatum*	*Oncidium crocidipterum*	*Polystachya cultriformis*
Pollen unit (LM)	Pollen unit	tetrad (tetrahedral, planar, decussate)	monad	tetrad (tetrahedral, planar, decussate)	tetrad (tetrahedral, planar, decussate)	tetrad (tetrahedral, planar, decussate)
	Dispersal unit	pollinium	pollinium	pollinium	pollinarium	pollinarium
	Aperture condition	inaperturate	ulcerate	inaperturate	inaperturate	inaperturate
Ornamentation (SEM)		psilate to fine striate to fossulate, perforate	foveolate to reticulate, heterobrochate	psilate, perforate to fossulate orgranulate	psilate, perforate	psilate
Tetrads of outer stratum 1 (TEM)	Layer 1	continuous, compact, sporopollenin	X	continuous, compact, sporopollenin	continuous, compact, sporopollenin	continuous, thick, sporopollenin
	Layer 2	thick, electron-translucent with electron-dense columellate to granulate elements	electron-translucent	thick, electron-translucent with electron-dense granules	thick, electron-translucent with electron-dense granules	electron-translucent with electron-dense granules
	Layer 3 (Intine)	thin, mono-layered	thin intine	monolayered	monolayered	bilayered
	Tectum	X	semitectate	X	X	X
	Infratectum	X	columellate	X	X	X
	Foot layer	X	thin, continuous	X	X	X
	Protoplast detachment	+	X	+	+	+
Tetrads of inner stratum 2 (TEM)	Layer 2	+	X	+ with loosely scattered granules	+ with loosely scattered granules	+
	Layer 3	+	X	+	+	+
	Protoplast detachment	+	X	+	+	X
Tetrads of inner stratum 3 (TEM)	Layer 2	+	X	+ very little or no granules	+with sparsely embedded granules	X
	Layer 3	+	X	+	+	X
	Protoplast detachment	X	X	X	+	X

Note: X—not present, +—present.

**Table 3 plants-13-01114-t003:** Staining methods for the chemical detection of pollen wall layers and various other pollen characteristics (miscellaneous), investigated using light and electron microscopy (LM and EM).

Species			*Bulbophyllum retusiusculum*	*Cephalanthera longifolia*	*Dendrobium* × *delicatum*	*Oncidium crocidipterum*	*Polystachya cultriformis*
Staining methods (TEM)	Potassium permanganate	Layer 1	electron-dense	electron-dense ektexine	electron-dense	electron-dense	electron-translucent
		Layer 2	electron-translucent	electron-translucent	electron-translucent	electron-translucent	electron-translucent
		Layer 3/Intine	electron-translucent	electron-translucent	electron-translucent	electron-translucent	electron-translucent
		Elements in layer 2	electron-dense	X	electron-dense	electron-dense	electron-translucent
	Thiéry test	Layer 1	n.i.	n.i.	n.i.	n.i.	X
		Layer 2	n.i.	n.i.	n.i.	n.i.	unclear
		Layer 3/Intine	n.i.	n.i.	n.i.	n.i.	++
		Elements in layer 2	n.i.	n.i.	n.i.	n.i.	X
Miscellaneous	Cellular condition (LM)	Pollen	two-celled	two-celled	two-celled	two-celled	two-celled
	Reserves in cytoplasm (LM)	Starch	X	X	X	X	X
		Lipids	+	X	+	X	+
	Sticky materials (LM, EM)	Pollen coatings/elastoviscin	little	little	plenty	plenty	little
	Appendages (LM, EM)	Frenicular caudicle	X	X	X	+	+
		Appendicular caudicle	X	X	X	X	X
		Stipe	X	X	X	+	+
		Viscidium	X	X	X	+	unclear
	Ubisch bodies (LM, EM)	Anther cap	n.i.	n.i.	X	n.i.	X

Note: X—not present, +—present, ++—positive reaction, n.i.—not investigated.

**Table 4 plants-13-01114-t004:** Investigated species: Hortus Botanicus Vindobonensis (HBV), cultivated in the Greenhouses of the Botanical Garden of Vienna; geocoordinates are listed in decimal degrees.

Clades	Tribe	Species	Location	ID
Higher epidendroids	*Cymbidieae*	*Oncidium crocidipterum*	HBV (greenhouse)	ORCH100464
Higher epidendroids	*Vandeae*	*Polystachya cultriformis*	HBV (greenhouse)	ORCH050167
Lower epidendroids	*Dendrobieae*	*Bulbophyllum retusiusculum*	HBV (greenhouse)	ORCH080268
Lower epidendroids	*Dendrobieae*	*Dendrobium* × *delicatum*	HBV (greenhouse)	ORCH800515
Lower epidendroids	*Neottieae*	*Cephalanthera longifolia*	Lobau (48.194859° 16.476873°)	CEPLON150522

## Data Availability

All data supporting the findings of this study are either part of the published manuscript or available from C.P. upon reasonable request.

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
