# Peer review of "Morphological and Ultrastructural Features of Selected Epidendroideae Pollen Dispersal Units and New Insights into Their Chemical Nature"

_plants, 2024, doi:10.3390/plants13081114_

Round 1

Reviewer 1 Report

Comments and Suggestions for Authors

The overall concept of the MS is good arranged well and falls within the scope. However, it still needs improvement to enhance its quality for possible publication in this journal.

Line 157-206: The authors provide both SEM and TEM results. Please separate the SEM results such as pollen sculpturing in one paragraph and the TEM findings such as ultrastructure of the pollen in a paragraph to make it easier for the readers.

Note: Please do it for all the species

In Table 1, please clarify and provide the polyad's length and width and then measure the individual grain diameter of the polyads. This is important for future studies about the phylogentic relationship of this subfamily with others.

In Table 2, Please confirm the exine sculpturing for the species Oncidium crocidipterum

Line 284: “and lumina of different diameter” hetrobrochate pollen right?

Line 285: “(nano to micro-verrucate, granulate)” why not verrucate to scabrate? Please revise this carefully.

“4.2.8. Scanning Electron Microscopy – surface sculpture and ornamentation” Please use only one surface sculpture or ornamentation.

For all the species please keep the format same for a better understanding of the results such as;

First, separate the SEM and TEM results.

Provide Size of the polyads (…… x ……µm).

Size of the individual grains such as polar axis and equatorial diameter (…… x ……µm).

The shape of the polyads and then the shape of the individual pollen.

Provide a P/E ratio based on the size of the pollen.

Exine sculpturing such as reticulate, scabrate, verrucate, perforate etc…

Please also measure the lumina diameter in case of the reticulate pollen.

Aperture membrane ornamentation,

Measure the aperture length and width.

Arrange and provide this information for all the investigated species.

Most importantly, please provide a taxonomic key based on the pollen traits to delimit the species.

The study is very interesting and will add significant contributions to the field. 

Comments on the Quality of English Language

 Minor editing of English language required

Author Response

Response to the comments and suggestions of reviewer 1:

Line 157-206: The authors provide both SEM and TEM results. Please separate the SEM results such as pollen sculpturing in one paragraph and the TEM findings such as ultrastructure of the pollen in a paragraph to make it easier for the readers.

Note: Please do it for all the species

This has been changed accordingly for all studied species.

In Table 1, please clarify and provide the polyad's length and width and then measure the individual grain diameter of the polyads. This is important for future studies about the phylogentic relationship of this subfamily with others.

All investigated species in the study have pollinia as pollen dispersal units and not polyads, which are typical for Fabaceae and not for orchids. For all the pollinia the size (length and width) is provided in the results and in table 1.

In Table 2, Please confirm the exine sculpturing for the species Oncidium crocidipterum

We changed the ornamentation from “psilate” to “psilate, perforate” in the table and also in the results.

Line 284: “and lumina of different diameter” hetrobrochate pollen right?

Yes, this is heterobrochate. We changed the text accordingly: “In SEM, pollen wall ornamentation in the interapertural area is foveolate to reticulate and heterobrochate, with thick muri.”

Line 285: “(nano to micro-verrucate, granulate)” why not verrucate to scabrate? Please revise this carefully.

The term scabrate is used for light microscopy only. As the description is based on SEM pictures, the terms “nano to micro-verrucate, granulate” are correct. The pollen terminology follows Halbritter et al. (2018).

“4.2.8. Scanning Electron Microscopy – surface sculpture and ornamentation” Please use only one surface sculpture or ornamentation.

This has been changed to ornamentation.

For all the species please keep the format same for a better understanding of the results such as;

First, separate the SEM and TEM results.

Has been separated.

Provide Size of the polyads (…… x ……µm).

All species investigated for this study have pollinia as pollen dispersal units and not polyads, which are typical for Fabaceae and not for orchids. For all the pollinia the size (length and width) is provided in the results and in table 1. Moreover, only one pollinium was measured, due to the limited plant material. We included this in the material and methods in chapter 4.2.7. Size measurements to make this clearer.

Size of the individual grains such as polar axis and equatorial diameter (…… x ……µm).

Individual pollen grains (monads) are only present in Cephalanthera longifolia, and for this species the size measurements and the shape are described in the results. Pollen size measurements have already been made for all species for the longest axis (=equatorial diameter). In this case, the measurement for the polar axis is not necessary.

The shape of the polyads and then the shape of the individual pollen.

As mentioned before there are no polyads in orchids, and the pollen dispersal unit is a pollinarium/pollinium. The description of the shape of the pollinarium/pollinium is added to the results. The shape of the tetrads and individual pollen grains is not relevant for orchids.

Provide a P/E ratio based on the size of the pollen.

The P/E ratio can be provided for LM (hydrated, acetolyzed), for hydrated pollen in SEM and also for dry pollen in SEM. The P/E ratio according to Halbritter et al. (2018) is described as “prolate”, “oblate” or “isodiametric”. Size measurements related to the ratio of the length of the polar axis (P) to the equatorial diameter (E) can be used if they make sense, which is not the case here.

Exine sculpturing such as reticulate, scabrate, verrucate, perforate etc…

The SEM results are now highlighted by a paragraph in the results.

Please also measure the lumina diameter in case of the reticulate pollen.

Has been added to the results.

Aperture membrane ornamentation, Measure the aperture length and width.

As the aperture is hidden inside the pollinarium, it is not possible to measure the aperture length and width.

Arrange and provide this information for all the investigated species.

We did our best to accommodate the requirements of the reviewer. Regarding the some of the measurements, it should be noted that pollen/tetrad measurements can be tricky. The size of pollen may vary due to natural variation within a single anther/flower or between anthers/flowers/populations etc. of a single taxon, and also depending on the preparation and methods used. For example, there is a considerable size difference between hydrated pollen observed with LM compared to SEM, as dehydration and critical point drying influence the state of pollen hydration. Therefore, we conclude that measurements, as performed in the literature (e.g., Damon et al. 2012), are not useful to delimit species by their size!

Damon, A.; Nieto L., G. Morphology of the Pollinia and Pollinaria of Orchids from Southeast Mexico. A Guide to the Morphology of the Pollinia and Pollinaria of Orchids from the Biological Corridor Tacaná-Boquerón in Southeast Mexico. Selbyana 2012, 31, 4–39, doi:10.2307/41760316.

Most importantly, please provide a taxonomic key based on the pollen traits to delimit the species.

The orchid family comprises between 24 and 30 thousand species in c. 736 genera. Even the subfamily under study has thousands of species. Therefore, we cannot provide a taxonomic key based on pollinia/tetrads/pollen that will be of any use since only a limited number of species have been investigated previously and for this study. Maybe in the future when “hundreds” of species have been investigated such key based on pollinia/tetrads/pollen might apply. Until then, scientist must use the flowers and macroscopic parts of the plants to segregate species.

Minor editing of English language required

The complete manuscript has been read over again and small linguistic errors as well as minor typos have been corrected. The English should now fulfil the journals requirements.

Reviewer 2 Report

Comments and Suggestions for Authors

It's an excellent paper on the morphology and ultrastructure of selected Epidendroideae pollen dispersal units and their chemical nature, based on detailed invesgation for five orchid species from same subfamily Epidendroideae. I'm happy to recommend this paper to be published in current form, and hopefully more investigations for representative species from another four subfamilies (Apostasioideae, Cypripedioideae, Orchidoideae Eaton, and Vanilloideae) could be done to compare their morphology and ultrastructure characteristics and chemical nature comprehensively.

Author Response

We would like to thank the referee for taking the time to review our manuscript.

Reviewer 3 Report

Comments and Suggestions for Authors

The manuscript contains very valuable and new palynological data on the Orchidaceae family, which will certainly initiate a broader scientific discussion on this topic. I consider it a great initiative by the authors. The work was prepared on a large scale, and various microscopic techniques were used for this research. This is the first such comprehensive study of orchid pollinaria, pollinia and pollen grains.

The introduction introduces the subject of the work very well and provides information that may be necessary for a reader less familiar with the topic. However, my most important reservation about the manuscript, which appears at the beginning of reading the text, is the lack of presented hypotheses. The manuscript in this version describes what would seem to be completely randomly selected species, without any justification for the choice of such species. In my opinion, authors absolutely should present their hypotheses and justify their choice of species to test them. This is the reason why I marked "average scientific soundness".

While I really liked the introduction to palynological issues, on the contrary, the flower structure diagram or the systematics of each of the studied species seems to be completely unnecessary at this level of detail in the manuscript. Due to the large number of illustrations, I would omit those showing the habit of the plants, because that is not what the work is about. One image of the flower per species would be enough.

Some of the remaining illustrations also raise my reservations. Many of them depict similar objects. I would suggest a more synthetic approach to presenting research documentation, which will certainly facilitate the reception of the entire text. I would also give up TEM figs of lower quality, with poor resolution and contrast. Sometimes it's better to stop at just the description when the photo doesn't show the details well anyway.

I presented minor comments directly in the text

Author Response

Response Letter to Reviewer’s comments:

We would like to thank you for your critical and useful comments.

We carefully revised the manuscript according to the comments.

Response to the comments and suggestions:

  1. The introduction introduces the subject of the work very well and provides information that may be necessary for a reader less familiar with the topic. However, my most important reservation about the manuscript, which appears at the beginning of reading the text, is the lack of presented hypotheses. The manuscript in this version describes what would seem to be completely randomly selected species, without any justification for the choice of such species. In my opinion, authors absolutely should present their hypotheses and justify their choice of species to test them. This is the reason why I marked "average scientific soundness".

We now include our hypotheses at the end of the introduction and justify the choice of species investigated for this study.

  1. While I really liked the introduction to palynological issues, on the contrary, the flower structure diagram or the systematics of each of the studied species seems to be completely unnecessary at this level of detail in the manuscript. Due to the large number of illustrations, I would omit those showing the habit of the plants, because that is not what the work is about. One image of the flower per species would be enough.

We agree with the reviewer on this point and reduced the systematics of each species, the amount of plant/flower illustrations to one flower illustration, and the flower illustration in the introduction.

  1. Some of the remaining illustrations also raise my reservations. Many of them depict similar objects. I would suggest a more synthetic approach to presenting research documentation, which will certainly facilitate the reception of the entire text. I would also give up TEM figs of lower quality, with poor resolution and contrast. Sometimes it's better to stop at just the description when the photo doesn't show the details well anyway.

We do not share the reviewer's opinion regarding the quality and resolution of the TEM pictures. All TEM pictures have a very high resolution and were made with a new TEM and high-resolution digital camera. As every plant material has a different contrast and staining behavior, the contrast can be different depending on the staining method used. In some TEM pictures the quality and/or contrast may look poor, but they are showing important stainings such as Lipid test, Thiéry test, or test for lipids with potassium permanganate, and are therefore important for the documentation and understanding of the results. As this was not a criticism of the other two reviewers, we will not change the TEM picture presentation and layout.